# Two-dimensional and multi-channel feature detection algorithm for the CALIPSO lidar measurements

Thibault Vaillant de Guélis[1,2], Mark A. Vaughan[3], David M. Winker[3], and Zhaoyan Liu[3]

[1]NASA Postdoctoral Program Fellow, NASA, Langley Research Center, Hampton, VA 23681, USA
[2]Science Systems and Applications, Inc., Hampton, VA 23666, USA
[3]NASA Langley Research Center, Hampton, VA 23681, USA

**Correspondence:** Thibault Vaillant de Guélis (thibault.vaillantdeguelis@outlook.com)

**Abstract.** In this paper we describe a new two-dimensional and multi-channel feature detection algorithm (2D-McDA) and demonstrate its application to lidar backscatter measurements from the Cloud-Aerosol Lidar and Infrared Pathfinder Satellite Observations (CALIPSO) mission. Unlike previous layer detection schemes, this context sensitive feature finder algorithm is applied to a 2D lidar "scene"; i.e., to the image formed by many successive lidar profiles. Features are identified when an extended and contiguous 2D region of enhanced backscatter signal rises significantly above the expected "clear air" value. Using an iterated 2D feature detection algorithm dramatically improves the fine details of feature shapes and can accurately identify previously undetected layers (e.g., subvisible cirrus) that are very thin vertically but horizontally persistent. Because the algorithm looks for contiguous 2D patterns using successively lower detection thresholds, it reports strongly scattering features separately from weakly scattering features thus potentially offering improved discrimination of juxtaposed cloud and aerosol layers. Moreover, the 2D detection algorithm uses the backscatter signals from all available channels: 532 nm parallel, 532 nm perpendicular, and 1064 nm total. Since the backscatter from some aerosol or cloud particle types can be more pronounced in one channel than another, simultaneously assessing the signals from all channels greatly improves the layer detection. For example, ice particles in subvisible cirrus strongly depolarize the lidar signal and, consequently, are easier to detect in the 532 nm perpendicular channel. Use of the 1064 nm channel greatly improves the detection of dense smoke layers, because smoke extinction at 532 nm is much larger than at 1064 nm, and hence the range-dependent reduction in lidar signals due to attenuation occurs much faster at 532 nm than at 1064 nm. Moreover, the photomultiplier tubes used at 532 nm are known to generate artifacts in an extended area below highly reflective liquid clouds, introducing false detections that artificially lower the apparent cloud base altitude, i.e. the cloud base when the cloud is transparent or the level of complete attenuation of the lidar signal when it is opaque. By adding the information available in the 1064 nm channel, this new algorithm can better identify the true apparent cloud base altitudes of such clouds.

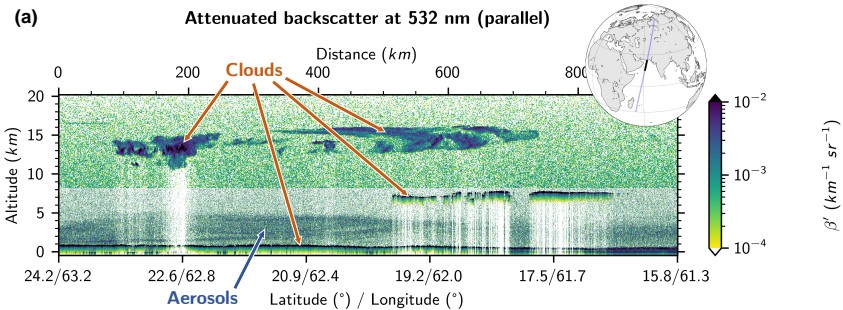

**Figure 1.** Curtain of lidar attenuated backscatter signal measured by CALIOP in the 532 nm parallel channel during nighttime observations on August 31, 2018, 21:46:37 UTC (start point), over Arabian Sea.

## 1  Introduction

The Cloud–Aerosol Lidar and Infrared Pathfinder Satellite Observation (CALIPSO) mission (Winker et al., 2010) has provided direct measurements of cloud and aerosol vertical distributions with a very high vertical resolution since 2006. A key com-
ponent of these measurements is made by the active remote sensing instrument CALIOP (i.e., the Cloud-Aerosol Lidar with Orthogonal Polarization), a two-wavelength (532 nm and 1064 nm), polarization sensitive elastic backscatter lidar.

The knowledge of the cloud and aerosol vertical distributions and their properties is critical in assessing the planet's radiation budget (e.g., Shonk and Hogan, 2010), in evaluating the atmospheric radiative heating rate (e.g., Huang et al., 2009), and for advancing our understanding of cloud-climate feedback cycles that occur as the climate warms (e.g., Tsushima et al., 2006).

The critically important first step in retrieving the spatial and optical properties of clouds and aerosols is to determine where these "features" are located in the vertical, curtain-like images (altitude vs. satellite track) of the backscattered lidar signals (Fig. 1). The CALIPSO feature detection algorithms were first developed for ground-based observations, and then adapted for space-based analyses using LITE measurements and CALIPSO simulations. These algorithms, which were conceived more than 25 years ago (e.g., Winker and Vaughan, 1994), at a time when computational power was considerably less than is now
available, are invoked sequentially on single, one-dimensional (1D) lidar signal profiles (possibly generated from averaging data from several consecutive laser pulses). Moreover, in order to minimize the computational load, the current CALIPSO algorithm is only applied to the 532 nm total signal (Vaughan et al., 2009).

To locate cloud and aerosol layers within lidar backscatter profiles, two main approaches are generally employed: the slope-based method, which looks for zero crossings in the first derivative of the raw signal (e.g., Pal et al., 1992) and threshold-based
methods, which search for regions exceeding some expectation of the maximum signal value that could be measured in "clear air" (e.g., Winker and Vaughan, 1994; Clothiaux et al., 1998; Campbell et al., 2008). Some studies use a combination of both methods (e.g., Wang and Sassen, 2001; Lewis et al., 2016). A few others adopt a third method: the wavelet analysis (e.g., Davis et al., 2000; Brooks, 2003). Because these layer detection algorithms are applied successively to individual 1D profiles (either single shot or averaged), we define them collectively as *profile-based processes*. We also define a second, more

comprehensive class of methods as *scene processes*. Scene processes can take advantage of the contextual information provided by a continuous series of profile measurements by searching for cloud and aerosol patterns in the two-dimensional (2D) image formed by successive lidar profiles. While edge detection techniques based on 2D gradient search routines are not well suited for spatial analysis of lidar data (Vaughan et al., 2005), methods based on sliding window operations have been shown to greatly improve the feature shape detection (e.g., Hagihara et al., 2010; van Zadelhoff et al., 2011; Herzfeld et al., 2014).

Here we propose a new 2D and multi-channel feature detection algorithm (2D-McDA). This "context sensitive" feature finder algorithm is then applied to a 2D lidar signal scene; i.e., to the image formed by many successive lidar profiles. Moreover, the 2D detection algorithm uses the backscatter signals from all available channels: the 532 nm co-polarized (or parallel) signal, the 532 nm cross-polarized (or perpendicular) signal, and the 1064 nm signal. Since the backscatter from some aerosol or cloud particle types can be more pronounced in one channel than another, simultaneously assessing the signals from all channels is expected to greatly improve the layer detection.

Section 2 presents a refined method for determining feature detection thresholds, which are a critically important component of the detection algorithm. Section 3 presents the detection algorithm. The detection of the Earth's surface is described first as it is performed first and separately from the cloud and aerosol detection. This has been shown to have many practical advantages. Then, the cloud and aerosol detection algorithm is described. Finally, the detections from each channel are merged into a composite feature detection mask. Section 4 shows how this new algorithm improves the feature detection compared to the CALIPSO version 4 vertical feature mask (VFM).

## 2 Threshold based feature detection

Atmospheric lidars measure attenuated signal backscattered by molecules ($m$) and particles ($p$)

$$\beta'(r) = (\beta_m(r) + \beta_p(r)) \, T_m(r)^2 T_{O_3}(r)^2 T_p(r)^2, \tag{1}$$

where $\beta_m(r)$ and $\beta_p(r)$ are the volume backscatter coefficients for molecules and particulates, and $T_m(r)^2$, $T_{O_3}(r)^2$ and $T_p(r)^2$ are, respectively, the two-way transmittances for molecules, ozone, and particles, and $r$ is the range from the satellite altitude. If there are no particles in the atmosphere, Eq. (1) reduces to the molecular attenuated backscatter coefficient

$$\beta'_m(r) = \beta_m(r) T_m(r)^2 T_{O_3}(r)^2. \tag{2}$$

A feature, i.e., a cloud or an aerosol layer, appears as an extended and contiguous region of enhanced attenuated backscatter signal that rises significantly above the expected clear sky (molecules only) value. However, not all signals that exceed the expected values of $\widehat{\beta'_m(r)}$ necessarily indicate the presence of features; instead, such excursions are often caused by noise. To distinguish features from the ambient (but noisy) clear sky signals, a first step is to determine a threshold above which signals can be confidently attributed to enhanced scattering arising from clouds or aerosols. We construct this threshold by first calculating the expected molecular attenuated backscatter, $\widehat{\beta'_m(r)}$, to which we add $k$ times the expected noise-induced standard deviation of the molecular signal. The resulting range-dependent threshold is the sum of $\widehat{\beta'_m(r)}$ and, based on error

propagation theory (e.g., Bevington and Robinson, 2003), $k$ times the root mean square (RMS) of the standard deviations due to both range-independent and range-dependent noise sources.

In constructing thresholds to be applied to CALIOP data, one must take into account the onboard signal averaging that is applied to the backscatter measurements. Because the CALIPSO satellite has limited telemetry bandwidth, the backscatter data is averaged both vertically and horizontally before the data is downlinked from the satellite, with increasing amounts of averaging applied to data acquired at higher altitudes (Hunt et al., 2009). As an example, signals acquired between 8.2 km and 20.2 km are averaged horizontally over three consecutive lidar pulses and vertically for four full resolution (15 m) range bins. Consequently, the downlinked profile data from within this region have been averaged over 12 full resolution onboard range bins. We compute a range-dependent threshold specifically tailored for the CALIOP profiles using

$$\beta'_T(r) = \widehat{\beta'_m(r)} + k\frac{f_{\mathrm{corr}}(r)}{\sqrt{N(r)}}\sqrt{\Delta\beta'_b(r)^2 + \widehat{\Delta\beta'_m(r)}^2},\tag{3}$$

where $\Delta\beta'_b(r)$ is the standard deviation due to background noise (range-independent[1]), $\widehat{\Delta\beta'_m(r)}$ is the expected standard deviation due to the shot noise (range-dependent) in the expected clear sky, and $N(r)$ is the number of bins averaged onboard. The $f_{\mathrm{corr}}(r)$ term is a correction function which takes into account the partial vertical correlation in samples due to the limited electronic bandwidth and the shifting and rebinning that can occur in the altitude registration phase of the level 1A processing (details in Appendix A). The number of shot noise standard deviations considered in the threshold is quantified by the factor $k$, which can be tuned according to the degree of sensitivity needed to avoid false detections. $\widehat{\beta'_m(r)}$ is derived from modeled profiles of molecular and ozone number densities. $\Delta\beta'_b(r)$ is derived from the on-board computation of the RMS of the background signal in the high altitude background region (HABR) between 65 and 80 km for each shot (Hostetler et al., 2006). $\widehat{\Delta\beta'_m(r)}$ is estimated using its proportional relation with the square root of $\widehat{\beta'_m(r)}$ (e.g., Liu and Sugimoto, 2002), called *noise scale factor* (NSF)

$$\widehat{\Delta\beta'_m(r)} = NSF_{\beta'}\sqrt{\widehat{\beta'_m(r)}}.\tag{4}$$

The NSF is evaluated from the solar background signal during daytime for the 532 nm parallel and perpendicular channels (Hostetler et al., 2006; Liu et al., 2006). At 1064 nm, CALIOP uses an avalanche photodiode (APD) detector, rather than the photomultiplier tubes (PMTs) that are used for the 532 nm channels. Because the APD dark noise overwhelms the 1064 nm shot noise, only the background noise is considered at 1064 nm.

Figure 2 shows the range-dependent threshold (red) computed from Eq. (3) with $k = 2$ applied to the 532 nm parallel lidar signal (blue) for a clear-sky case study during nighttime. Note the noise due to the quantum nature of photons (shot-noise) in this figure. Indeed, although background noise, mainly due to solar radiation, is quite low during nighttime, the lidar signal shows large variations around the expected clear-sky return (black). The range-dependent threshold correctly keeps most the signal below the detection level. Jumps at -0.5 km, 8.2 km, and 20.2 km reveal the change of onboard averaging resolution. However, a few points of the lidar signal still exceed the threshold. Some continuity tests are then needed to determine whether

---

[1]The background noise is range-independent in the digitizer-reading domain $P$. However, it then depends on $r$ when transformed to the $\beta'$ domain.

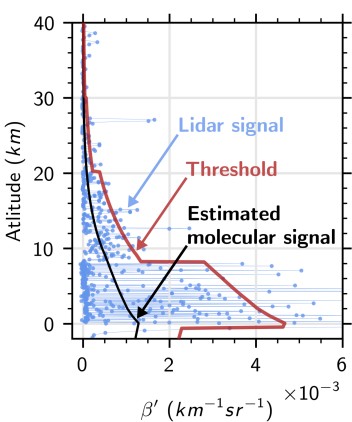

**Figure 2.** Range-dependent threshold (red) applied to a single shot lidar signal profile (blue) in clear-sky during nighttime. The estimated molecular signal is shown in black. Jumps in the lidar signal and threshold at -0.5 km, 8.2 km, and 20.2 km reveal the change of onboard averaging resolution.

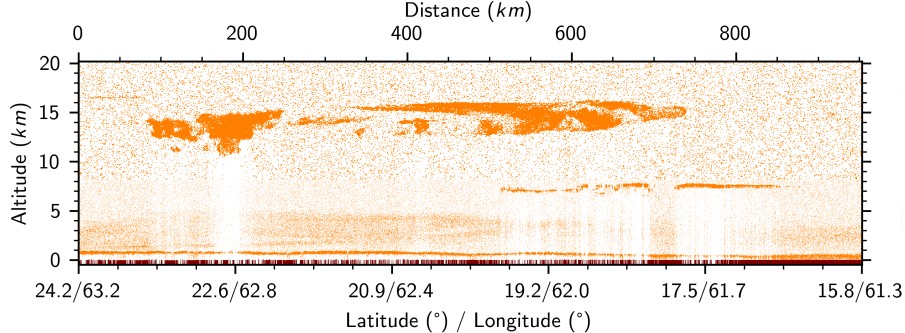

**Figure 3.** Pixels of Fig. 1 where the lidar signal is above the range-dependent threshold computed from Eq. (3) with $k = 2$ are shown in orange. Brown pixels show surface detection.

these high signals are due to noise or instead part of an extended feature. Unlike the current CALIPSO detection algorithm, this continuity test will be applied in two dimensions. Figure 3 shows all pixels of Fig. 1 where the lidar signal is above the range-dependent threshold computed from Eq. (3) with $k = 2$.

Like the current CALIOP layer detection algorithm, the 2D-McDA is applied to profiles of attenuated scattering ratios, defined as

$$R'(r) = \frac{\beta'(r)}{\beta'_m(r)}. \tag{5}$$

The attenuated scattering ratio threshold is then obtained from

$$
\begin{aligned}
R'_T(r) &= \frac{\beta'_T(r)}{\widehat{\beta'_m(r)}} \\
&= 1 + k \frac{f_{\mathrm{corr}}(r)}{\sqrt{N(r)}} \sqrt{\frac{1}{\widehat{\beta'_m(r)}^2} \Delta \beta'_b(r)^2 + NSF^2_{\beta'} \frac{1}{\widehat{\beta'_m(r)}}}.
\end{aligned}
\tag{6}
$$

Equation (6) is applied to the three lidar channels (532 nm parallel, 532 nm perpendicular, and 1064 nm) during the 2D-McDA process.

## 3   2D and multi-channel feature detection algorithm

2D-McDA is applied to the scattering ratio signals at 532 nm parallel, 532 nm perpendicular, and 1064 nm. First, the detection of the surface altitude is performed and the signal from this altitude and below is removed from the data. Second, the detection of cloud and aerosol layers is done in each channel based on iterated detection thresholds and 2D spatial continuity tests. Finally, the masks from the three channels are merged in a composite feature mask.

### 3.1   Surface detection

Before detecting cloud and aerosol layers using the detection threshold as described in the previous section, we perform first an independent detection of the Earth's surface. Doing the surface detection in a first and separate step allows a better retrieval of the surface echo and prevents complications in the cloud and aerosol layer detection process. Also, knowing where the surface is detected allows subsequent separation of *semi-transparent* features from *opaque* features, which is essential for accurately estimating range-resolved profiles of extinction coefficients (Young et al., 2018). Operationally, atmospheric features are defined as being opaque when no surface return or other atmospheric feature can be detected below them. From this definition it follows that the signals received from beneath opaque features have been fully attenuated within these features. The Earth surface detection algorithm used here is closely akin to the one described in Vaughan et al. (in progress) and is applied to the 532 nm parallel and 1064 nm channels (details in Appendix B). The signals from the top of the detected surface echo and below this point are removed from the data. To minimize computation times, the surface detection algorithm is not applied to the 532 nm perpendicular channel signal. The backscatter from ocean surfaces (covering ~70% of the planet) does not depolarize and, excluding snow and ice, the depolarization of most land surfaces is relatively low (Lu et al., 2017), hence the preponderance of the surface backscatter is in the parallel channel. The altitude retrieved in the parallel channel is used to remove signal at and below the estimated surface altitude in the perpendicular channel. Note that there is some small chance that a surface echo can appear in the perpendicular channel but not be visible/detected in the parallel channel. The detection of the surface corresponding to Fig. 1 is shown in brown on Fig. 3.

### 3.2   Cloud and aerosol detection

The detection of cloud and aerosol layers in a single channel curtain of lidar measurements takes place in four main steps:

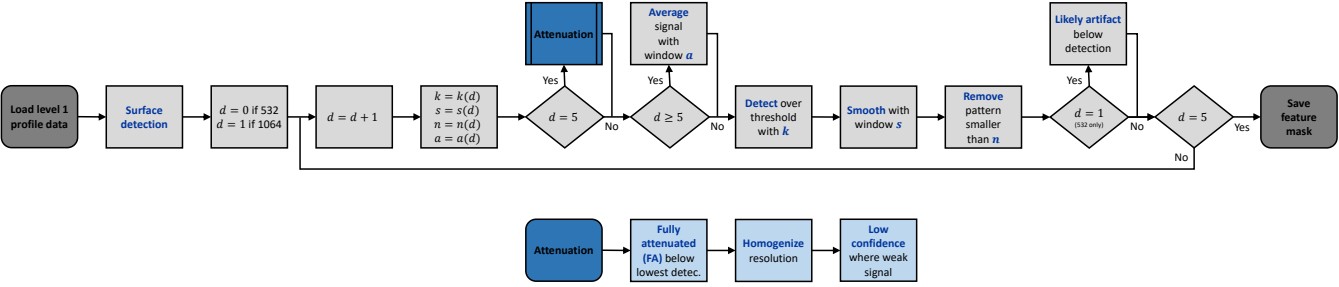

**Figure 4.** Flowchart of the two-dimensional and multi-channel feature detection algorithm (2D-McDA). $d$ is the detection step, $k$ is the number of total noise standard deviations used in the detection threshold Eq. (3), $s$ is the size of the window used for the spatial coherence test, $n$ is the minimum number of pixels in each pattern, and $a$ is the size of the averaging window. See the algorithm description in Sect. 3 and the coefficient values in Table 1.

1. Detecting strong features; i.e., identifying contiguous regions of enhanced attenuated scattering ratios that rise above the feature detection threshold (which is repeatedly decreased from very large value of $k$ down to $k = 1$) without applying any signal averaging (i.e., $d = 1 - 4$ in Table 1, Sect. 3.2.1);

2. Flagging regions below opaque features as "fully attenuated" (FA) and regions below transparent features where the signal is strongly attenuated with the low confidence flag "almost fully attenuated" (AFA) (Sect. 3.2.2);

3. Averaging of those signals not already flagged using a horizontal sliding window (Sect. 3.2.3);

4. Detecting faint features; features are once again identified as contiguous regions of enhanced signal (i.e., averaged attenuated scattering ratios) that rise above the recomputed feature detection threshold (Sect. 3.2.1).

Figure 4 shows the flowchart of the whole detection algorithm. The parameter values used at the different detection levels are given in Table 1.

The following subsections give the details of the main steps presented above.

### 3.2.1 Detection

The detection phase is performed following three substeps:

1. All pixels of within image that exceed the threshold are first flagged as detected (Fig. 3);

2. A spatial coherence test window is applied to the image of detected/undetected pixels. It smooths the shape of detected pattern and remove isolated noisy detected pixels by turning some of detected pixels to undetected or undetected to detected;

3. Smoothed patterns are required to meet a minimum numeric threshold of contiguous pixels. Patterns that fail to meet this threshold are removed from consideration for this level of detection.

**Table 1.** Coefficient $k$ in threshold detection, spatial coherence test window size $s$, minimum number of pixels in pattern $n$, and averaging window size $a$ used at each detection level $d$. Window sizes are given in vertical × horizontal in pixel number with pixel resolution of 30 m × 0.33 km.

| $d$ | | 1 | 2 | 3 | 4 | 5 |
|---|---|---|---|---|---|---|
| | $k$ | 100 | 20 | 2 | 1 | 1 |
| **532 nm parallel** | $s$ | – | – | 11×11 (330 m × 3.67 km) | 3×21 (90 m × 7 km) | 9×51 (270 m × 17 km) |
| | $n$ | 1 | 1 | 60 | 200 | 10000 |
| | $a$ | – | – | – | – | 1×15 (30 m × 5 km) |
| | $k$ | 500 | 100 | 2 | 1 | 1 |
| **532 nm perpendicular** | $s$ | – | – | 11×11 (330 m × 3.67 km) | 3×21 (90 m × 7 km) | 9×51 (270 m × 17 km) |
| | $n$ | 1 | 1 | 60 | 200 | 1000 |
| | $a$ | – | – | – | – | 1×15 (30 m × 5 km) |
| | $k$ | – | 20 | 2 | 1 | 1 |
| **1064 nm** | $s$ | – | – | 11×11 (330 m × 3.67 km) | 3×21 (90 m × 7 km) | 9×51 (270 m × 17 km) |
| | $n$ | – | 1 | 60 | 200 | 10000 |
| | $a$ | – | – | – | – | 1×15 (30 m × 5 km) |

The scattering ratio image used in the layer detection scheme has a spatial resolution of one laser pulse horizontally and 30 m vertically, equivalent to the finest spatial resolution of the CALIOP data. As described in Hunt et al. (2009), CALIOP data is averaged onboard the satellite with spatial resolutions that vary according to altitude. Scattering ratios in regions where the data resolution is coarser than the image resolution (30 m × 1/3 km horizontally) are duplicated as necessary to match the image resolution. For example, between 8.2 and 20.2 km, the spatial resolution of the signal is 1 km horizontally × 60 m vertically. These values are replicated 12 times to populate the corresponding area in the 30 m × 1/3 km scattering ratio image. The first substep is then to flag all pixels of this image which exceed the detection threshold given by Eq. (6) with the value of

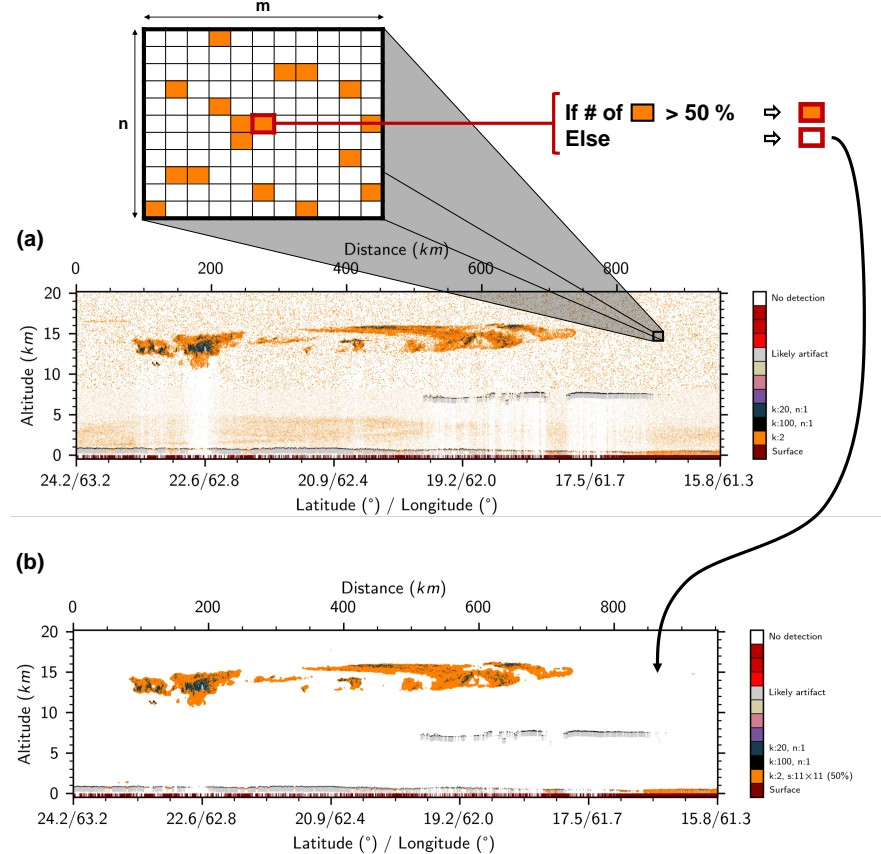

**Figure 5.** Illustration of the spatial coherence test window substep of 2D-McDA on the 532 nm parallel channel for the case study shown in Fig. 1. (a) Pixels which are over the detection threshold given by Eq. (6) with the value of $k = 2$ (orange). (b) Result after applying a $11\times11$ spatial coherence test window on the detected pixels. Note that the insert on the top is just an illustration and does not show the real content of the image portion.

$k$ defined in Table 1. For example, for the 532 nm parallel channel, at the detection level $d = 3$, the detected pixels are those where signal is greater than $1 + k$ times the expected noise standard deviation with $k = 2$ (Fig. 5a; orange pixels). Then, the second substep is to apply a spatial coherence test window (rows $a$ in Table 1) on these detected pixels (Fig. 5a,b). Here, an

$11\times11$ pixels window is applied to each pixel of the image, with the window being centered successively on all pixels. If the number of originally detected pixels in the window is greater than half of the total number of pixel in the window ($\geq 61$ for a $11\times11$ pixels window), then the center pixel is considered as detected. If not, the center pixel is considered as not detected. In this smoothing step, the determination of detection status does not rely on a single pixel exceeding its threshold, but instead on the fraction of neighboring pixels that exceed their thresholds. Consequently, a pixel classified as detected may not itself

exceed the detection threshold. Similarly, a pixel that exceeds the threshold may not ultimately be classified as detected. The

pixel count within the window is limited to those detected at the current detection level $d$ and at the previous detection level $d-1$. This allows detection continuity of similar backscatter intensities and avoids connecting noise encountered during fainter detections to a strong feature detected earlier. Other flagged pixels (i.e., "Surface", detection $\leq d-2$, "Likely artifact", "Fully attenuated", "Almost fully attenuated", and "Low confidence small strips" (to be described in detail later in this subsection and subsection 3.2.2) and pixels outside the window when at the top or bottom edges of the image are not considered in the window and the total number of candidate pixels in the window is decreased accordingly. The shapes of detected features are smoothed by the spatial coherence test window, while the noise (isolated orange pixel) is removed. However, some small clusters of pixels sometimes persist. Those small clusters cannot be confidently declared as feature at this stage. They can be due to noise or they can be part of a larger, fainter feature. Then, we decide not to consider these small patterns as detected feature and retain these regions for inclusion in the signal averaging used in successive iterations of the algorithm. To be declared as detected feature, smoothed detected patterns need to consist of more than $n$ connected pixels (see Table 1), otherwise they are removed from this detection level.

This detection procedure is applied several times (the successive detection level $d$ of Table 1) with different thresholds, different spatial coherence test window $s$, and different limits on the number of connected pixels required $n$ (Table 1) in order to detect all layers from the most evident, very strong patterns to the very faint ones, and from geometrically small patterns to very extended ones. Note that the horizontal spatial coherence test window (3×21) enables the detection of faint but horizontally extended cirrus such as the layer shown between 50 km and 100 km in Fig. 1. The detection of this subvisible cirrus is presented in Fig. 6. Figure 6a to Fig. 6b shows the implementation of the 3×21 spatial coherence test window. We see that the cirrus pattern is smoothed and now clearly appears on Fig. 6b due to the fact that most of the noise around has been removed. However, many small clusters of noise pixels persist. By applying the minimum numeric threshold of connected pixel $n$ on the detected pattern, we are able to remove small cluster due to noise while keeping the real cirrus (Fig. 6c).

### 3.2.2 Special flags

For the 532 nm channels, a first detection of very strong signal is performed (see $d = 1$ in Table 1). The aim of this initial scan is to identify the tops of very strongly scattering liquid clouds and ice clouds containing high fractions of horizontally oriented ice (HOI) crystals. The non-ideal transient response by PMTs following these very strong signals often generates a spurious, exponentially decaying signal enhancement in the underlying range bins (McGill et al., 2007; Hunt et al., 2009; Lu et al., 2020). The presence of these "noise tails" in the 532 nm signals can introduce large biases into the determination of the apparent bases of opaque water clouds. To exclude this artifact in the detection process, the 600 m below the base of the detected very strong signal are flagged as "Likely artifact" and removed from the signal. Since the APD used in 1064 nm channel does not produce these noise tails, we rely on the 1064 nm channel for the detection of the apparent base of these strongly scattering layers.

After detection of the strongest features, i.e without signal averaging ($d = 1-4$ in Table 1), we flag as "Fully attenuated" (FA) all pixels below a detected strong feature where the surface has not been detected. In this portion of the profile, the signal is too weak to be further exploited. Second, the contiguous pixels located in the vertical extent between two detected features are flagged as "Almost fully attenuated" (AFA) whenever the backscatter intensity falls below an empirically determined threshold.

For the 532 nm parallel channel, these pixels are flagged as AFA when more than 30 % of the population has backscatter intensities that are less than 10 % of the corresponding detection thresholds. These pixels are flagged AFA in the 532 nm perpendicular channel whenever the signals in more than 90 % of the population fall below (100 % of) the corresponding threshold values. To be flagged as AFA in the 1064 nm channel, more than 85 % of the population must have signal less than (100 %) of the corresponding threshold. In all cases, the AFA thresholds were determined experimentally and are tunable.

Finally, the horizontal distance between successive (A)FA columns can be very small and the likelihood of confidently detecting features in these narrow gaps is very low. For this reason, the data in all horizontal extents smaller than 5 km (15 profiles) that lie between (A)FA columns are flagged as "Low confidence small strips".

After removing all data identified with these low confidence flags from the attenuated scattering ratios, the signal is averaged in order to try to detect fainter features.

### 3.2.3 Signal averaging

We then average the remaining signal (here the attenuated scattering ratios) using a Gaussian sliding window that extends over 5 km (15 profiles) horizontally and a single range bin vertically ($a$ in Table 1). Using a sliding window, instead of the fixed window used in the CALIOP feature detection algorithm, provides much improved resolution of the horizontal edges position of faint features (1/3 km instead of 5, 20, or 80 km) and makes it possible to detect non-uniform horizontal edges. A Gaussian weight with a standard deviation of 1.67 km is applied, thus giving a stronger weight to pixels closer to center of the window than at the edges. We chose a horizontal window here because the spatial extent of very faint layers is mainly in the horizontal direction. Typically, thin cirrus have geometrical thicknesses of a few hundreds of meters but spread horizontally over several hundreds of kilometers. The use of a horizontal averaging window thus allows the detection of thin layers close to each other vertically. Pixels flagged as surfaces or features are not considered in the averaging window. However, if the center pixel of the averaging window (i.e. the pixel to which the averaging is applied) is a low confidence pixel (i.e. "Likely artifact", (A)FA, or "Low confidence small strips"), then the averaging window is applied and, if the average signal value exceeds the detection threshold, this center pixel in the feature detection mask is "unflagged" until the end of the detection level processing, after which its low confidence flag is restored. This allows us to maintain connections between features separated by a few low confidence pixels. Once the averaging is performed, the detection substeps (Sect. 3.2.1) are then applied to the averaged signal. Note too that horizontally adjacent features separated only by a low confidence vertical band (i.e., pixels classified as FA, AFA, and/or small strips) are considered as a single, merged feature when counting the number of connected pixels. Some examples of this horizontal merging are seen in the smaller fragments of the aerosol layer found at about 4 km and an along track distance of 500 km to 750 km in Fig. 7.

Figure 7 shows the final mask for the 532 nm parallel channel after the detection of the faint features.

### 3.3 Three channels composite detection

The detection algorithm is applied individually to the lidar signal from each of the three channels (Fig. 8) and all pixels identified as features in any of the three channels are retained in the composite mask (Fig. 9a). Comparing this new feature

mask (Fig. 9a,b) to the current version of the Vertical Feature Mask (VFM) (Fig. 9c), we first note the improvement in the detected contour of the large cirrus. We also note that the 2D-McDA readily detects faint cirrus (e.g., as seen between 0 km and
75 km) that is missed by the current VFM. The vertical spreading of the clouds seen in the VFM at around 7.5 km in altitude and between 500 km and 900 km horizontally is due to the aforementioned PMT artifact afflicting the 532 nm signals beneath strongly scattering layers. This is not seen in the 2D-McDA feature mask because pixels below the cloud top are flagged as "Likely artifact" in the 532 nm channels and so we make no attempt to retrieve the cloud apparent base of such opaque clouds at this wavelength. Instead, in these cases we retrieve the true penetration depth estimates using the 1064 nm signals (Fig. 8c),
which are not affected by detector transient response artifacts (see light blue = "1064 only").

## 4    Performance assessments and comparisons to version 4

In this section, we present two case studies to show the improvements bring by this new feature detection approach.

### 4.1    Variety of cloud type and shape

Figure 10 presents the attenuated backscattered lidar signal in the three channels for another case study showing a variety of
cloud types and shapes which occurred above Ethiopia on August 31, 2018 during nighttime. We can see that the artifacts below liquid water clouds (close to the surface and up to 8 km) appear in the 532 nm parallel (Fig. 10a) and the 532 nm perpendicular (Fig. 10b) channels but not at 1064 nm (Fig. 10c). We note that thin cirrus clouds, like the one at 17 km in altitude between 1550 km and 1850 km, are clearly brought out in the 532 nm perpendicular channel (Fig. 10b). If we look now at the composite feature detections derived from these three signals (Fig. 11a), we note again how well the apparent bases of liquid clouds are
retrieved by using the 1064 nm channel. We note also that the successful identification of thin cirrus can largely be attributed to our use of the 532 nm perpendicular channel. Figure 11b shows the same mask as Fig. 11a but with the same colors that are used for the VFM images (Fig. 11c). This change of colors is intended to facilitate one-to-one comparisons between the two detection schemes. However, note that the yellow and white colors do not discriminate aerosol from cloud, as in the VFM, but instead simply differentiate weak from strong features based on whether the feature detection required data averaging (yellow)
or not (white). Finally, Figure 11d shows the difference between the new composite feature detection mask and the VFM. We see that the contour of features retrieved by the 2D-McDA represents a distinct improvement over the squared boundaries reported by the VFM. We note too that the new algorithm detects thin clouds that are obviously missed by the VFM and that it eliminates significant detection artifacts reported by the VFM between 700 km and 900 km.

### 4.2    Dense smoke

Figure 12 presents a dense smoke event in Siberia on July 26, 2006 during daytime. The smoke layer is opaque at 532 nm and thus we do not see any surface echo for this channel (Fig. 12a). Note that the smoke is non-depolarizing so there is no perpendicular signal (Fig. 12b). Because the standard CALIOP layer detection only examines the 532 nm channel, the VFM (Fig. 13c) indicates that the signals are fully attenuated after detecting (at 532 nm) the apparent base of the smoke layer.

However, at 1064 nm the dense smoke layer is semi-transparent because the 1064 nm signals are attenuated significantly less than at 532 nm. Then, the surface is readily detected at 1064 nm (Fig. 12c). This scene clearly illustrates the advantage gained by using a multi-channel feature detection algorithm, since the full vertical extent of the smoke plume can only be retrieved by inspecting the 1064 nm measurements (light blue in Fig. 13a).

## 5 Conclusions

This paper describes the architecture and theoretical underpinnings of a new two-dimensional, multi-channel feature detection algorithm (2D-McDA) used to identify layer boundaries in the backscatter signals acquired by the elastic backscatter lidar aboard the Cloud Aerosol Lidar and Infrared Pathfinder Satellite Observations (CALIPSO) platform. The cloud and aerosol layer detection boundaries reported in the standard CALIPSO data products are detected by scanning sequences of 532 nm attenuated scattering ratio profiles constructed at increasingly coarser horizontal averaging resolutions. In contrast, the 2D-McDA is more akin to an image processing algorithm that examines full resolution lidar scenes and hence can identify many of the fine details that are often obscured by CALIPSO's standard multi-resolution averaging scheme. Relative to the CALIPSO version 4.2 vertical feature mask (VFM) data product, the 2D-McDA shows the following improvements.

- Because it is applied to single profiles averaged over several different horizontal resolutions, the standard CALIOP feature detection produces blocky, rectangular layers. The complex shapes of aerosol and cloud features are better preserved by the 2D-McDA windowing and data aggregation operations, which provide the flexibility required to distinguish fine spatial details. It is hoped that this improved feature detection will lead to improvements in classifying features according to type (e.g., clouds vs. aerosols) and in their optical property retrievals. Ideally, separate identification of strongly scattering and weakly scattering features by the 2D-McDA will also offer improved discrimination between juxtaposed cloud and aerosol layers or identifiable regions of ice and liquid water within a cloud. The improvement of the cloud shape detection is by itself important for example for studies interested in anvil clouds (e.g., Bony et al., 2016; Hartmann, 2016);

- The detection of subvisible cirrus is significantly enhanced by both the 2D detection scheme and the use of the 532 nm perpendicular channel, which is especially sensitive to the presence of depolarizing ice crystals. Those clouds play an important role in the climate system as they regulate the vertical transport of water vapor near the upper troposphere–lower stratosphere (e.g., Jensen et al., 1996; Luo et al., 2003), influence the local thermal budget, and drive dynamics of the tropopause region (e.g., Hartmann et al., 2001; McFarquhar et al., 2000);

- The apparent base altitudes of highly reflective clouds, i.e. the levels of complete attenuation of the lidar signal, which are routinely biased low (by several hundred meters) due to the non-ideal transient response of 532 nm photomultiplier tubes, are now more correctly retrieved by incorporating measurements made by the 1064 nm channel. The apparent cloud base altitude, which results from both attenuation of the direct beam and multiple scattering effects, has been directly linked with the amount of longwave radiation escaping the Earth at the top of the atmosphere (Vaillant de Guélis

et al., 2017a, b), making its accurate estimation very important for cloud feedback studies (Vaillant de Guélis et al., 2018);

- The 2D-McDA can retrieve the full vertical extent of dense smoke layers by examining the 1064 nm channel. Within smokes, the 1064 nm signals are attenuated significantly less than at 532 nm and hence can more often penetrate the full vertical extent of these layers. Those biomass burning aerosols play a significant role in the Earth's radiative balance by their scattering and absorption of incoming solar radiation (e.g., Penner et al., 1992; Christopher et al., 1996) and the interaction they have with clouds (e.g., Kaufman and Fraser, 1997). The full detection of those layers will lead to more accurate aerosol optical depth retrievals which will improve estimates of the radiation budget. Profiling the full depth of the smoke layer will also help to understand whether the layer is in contact with underlying clouds and able to affect cloud microphysics.

While the current implementation of the algorithm is computationally intensive, numerous optimizations are underway, and it is now feasible to apply the 2D-McDA operationally using CALIPSO's available computer resources. However, while fundamentally important, feature detection is only the first step in extracting a comprehensive suite of geophysical parameters from raw lidar measurements. Taking full advantage of the improved spatial analyses delivered by the 2D-McDA thus requires the development of a companion set of 2D scene processes to replace the 1D profile-based processes that are currently used in the CALIPSO retrieval architecture to perform the essential tasks of discriminating between clouds and aerosols, identifying cloud thermodynamic phase, and classifying aerosols by type.

*Data availability.* The CALIPSO level 1 lidar profile product used throughout this study is publicly distributed by the NASA Langley Research Center Atmospheric Science Data Center (Vaughan et al., 2018, NASA Langley Research Center Atmospheric Science Data Center; https://doi.org/10.5067/CALIOP/CALIPSO/LID_L1-Standard-V4-10; last access: 18 December 2020).

## Appendix A: Correction functions

### A1 Correction due to electronic bandwidth

A correction should be applied to Eq. (3) due to the fact that the nominal sample range interval (15 m) of the lidar is smaller than its range resolution ($\approx 40$ m) determined by the electronic bandwidth (2 MHz; Hunt et al., 2009). Consequently, a 15-m sample bin is partially correlated with the two bins above and the two bins below. As a result, vertical averaging of several 15-m bins $N_{\mathrm{bin}}$ does not reduce the noise standard deviation as much as it would if the samples were independent. A function $f_a(N_{\mathrm{bin}}) > 1$ is then applied to correct from this partial correlation. This function is evaluated as follows.

A 15-m bin $b_i$ has a variance $\text{Var}(b_i)$. If $N_{\text{bin}}$ 15-m bins are vertically averaged together to form a larger bin $B$, then the variance of the mean is given by

$$\text{Var}(B) = \text{Var}\left(\frac{1}{N_{\text{bin}}} \sum_{i=1}^{N_{\text{bin}}} b_i\right) = \frac{1}{N_{\text{bin}}^2} \text{Var}\left(\sum_{i=1}^{N_{\text{bin}}} b_i\right)$$

$$= \frac{1}{N_{\text{bin}}^2} \sum_{i,j=1}^{N_{\text{bin}}} \text{Cov}(b_i, b_j) \tag{A1}$$

$$= \frac{1}{N_{\text{bin}}^2} \left( \sum_{i=1}^{N_{\text{bin}}} \text{Var}(b_i) + 2 \sum_{1 \le i < j \le N_{\text{bin}}} \text{Cov}(b_i, b_j) \right),$$

where $\text{Cov}(\cdot, \cdot)$ is the covariance. When averaging $N_{\text{bin}}$ consecutive 15-m bins, we can consider they have approximately the same range and then that their variance is constant: $\text{Var}(b_i) = \text{Var}(b)$. If the $b_i$ bins were uncorrelated, then we would have $\text{Cov}(b_i, b_j) = 0$, $\forall (i \ne j)$, and then $\text{Var}(B) = \frac{\text{Var}(b)}{N_{\text{bin}}}$. However, since each bin is partially correlated with its vertical neighbors, we have $\text{Cov}(b_i, b_{i+m}) = \text{constant}$ $\forall i$ for each lag of $m$ range bins. Then, Eq. (A1) can be rewritten following

$$\text{Var}(B) = \frac{\text{Var}(b)}{N_{\text{bin}}} \left( 1 + \frac{2}{N_{\text{bin}}} \sum_{1 \le i < j \le N_{\text{bin}}} \frac{\text{Cov}(b_i, b_j)}{\text{Var}(b)} \right)$$

$$= \frac{\text{Var}(b)}{N_{\text{bin}}} \left( 1 + 2 \sum_{m=1}^{N_{\text{bin}}-1} \frac{N_{\text{bin}} - m}{N_{\text{bin}}} R(m) \right), \tag{A2}$$

where $R(m) = \frac{\text{Cov}(b_i, b_{i+m})}{\text{Var}(b)}$ is the autocorrelation coefficient for a lag of $m$ range bins. It follows that the correction function to apply on the total noise standard deviation in Eq. (3) to take into account the vertical partial correlation due to the electronic bandwidth is

$$f_a(N_{\text{bin}}) = \left( 1 + 2 \sum_{m=1}^{N_{\text{bin}}-1} \frac{N_{\text{bin}} - m}{N_{\text{bin}}} R(m) \right)^{1/2}. \tag{A3}$$

## A2 Correction due to redistribution in altitude registration

An additional correlation arises from the data redistribution in the altitude registration of level 0 data during the level 1A processing. Indeed, the altitude of the sample bins of a raw data profile are recalculated with more accurate information about the satellite altitude and laser viewing angle in the data processing on ground. A shift for a few range bins (no more than three in most of the cases) can be needed for the full resolution (30 m) samples. The number of 30-m bins shifted $N_{\text{shift30}}$ (which we express below in terms of an equivalent number of 15-m bins shifted $N_{\text{shift15}}$) only add correlation to regions in the profile data where the vertical range resolution is coarser than 30 m, i.e. where the vertical range resolution is 60 m, 180 m, and 300 m (Winker et al., 2006). Indeed, in those regions, a vertical shift by $N_{\text{shift30}}$ 30-m bins lead to the necessity of rebinning two neighboring bins larger than 30 m which introduce additional correlation to those bins. When there is a shift of $N_{\text{shift15}}$ 15-m bins (an even number since shifts are performed at 30 m resolution), each new shifted bin $B'_k$, with vertical resolution coarser

than 30 m, is computed from the weighted average of the two original bins (with $N_{\text{bin}}$ size resolution) it steps across $B_k$ and $B_{k+1}$ (Fig. A1) following

$$B_k' = \frac{N_{\text{bin}} - N_{\text{shift15}}}{N_{\text{bin}}} B_k + \frac{N_{\text{shift15}}}{N_{\text{bin}}} B_{k+1}, \tag{A4}$$

where

$$B_k = \frac{b_1 + b_2 + \cdots + b_{N_{\text{bin}}}}{N_{\text{bin}}} = \frac{1}{N_{\text{bin}}} \sum_{i=1}^{N_{\text{bin}}} b_i, \tag{A5}$$

and

$$B_{k+1} = \frac{b_{N_{\text{bin}}+1} + b_{N_{\text{bin}}+2} + \cdots + b_{N_{\text{bin}}+N_{\text{bin}}}}{N_{\text{bin}}} = \frac{1}{N_{\text{bin}}} \sum_{j=1}^{N_{\text{bin}}} b_{N_{\text{bin}}+j}. \tag{A6}$$

The variance of a $B_k'$ can be written

$$
\begin{aligned}
\text{Var}(B_k') &= \text{Var}\left( \frac{N_{\text{bin}} - N_{\text{shift15}}}{N_{\text{bin}}} B_k + \frac{N_{\text{shift15}}}{N_{\text{bin}}} B_{k+1} \right) \\
&= \left( \frac{N_{\text{bin}} - N_{\text{shift15}}}{N_{\text{bin}}} \right)^2 \text{Var}(B_k) + \left( \frac{N_{\text{shift15}}}{N_{\text{bin}}} \right)^2 \text{Var}(B_{k+1}) + 2 \frac{N_{\text{bin}} - N_{\text{shift15}}}{N_{\text{bin}}} \frac{N_{\text{shift15}}}{N_{\text{bin}}} \text{Cov}(B_k, B_{k+1}) \\
&= \left[ \left( \frac{N_{\text{bin}} - N_{\text{shift15}}}{N_{\text{bin}}} \right)^2 + \left( \frac{N_{\text{shift15}}}{N_{\text{bin}}} \right)^2 \right] \frac{\text{Var}(b)}{N_{\text{bin}}} f_a(N_{\text{bin}})^2 + 2 \frac{N_{\text{bin}} - N_{\text{shift15}}}{N_{\text{bin}}} \frac{N_{\text{shift15}}}{N_{\text{bin}}} \frac{\text{Var}(b)}{N_{\text{bin}}^2} \sum_{i,j=1}^{N_{\text{bin}}} \frac{\text{Cov}(b_i, b_{N_{\text{bin}}+j})}{\text{Var}(b)} \\
&= \frac{\text{Var}(b)}{N_{\text{bin}}} \left\{ \left[ \left( \frac{N_{\text{bin}} - N_{\text{shift15}}}{N_{\text{bin}}} \right)^2 + \left( \frac{N_{\text{shift15}}}{N_{\text{bin}}} \right)^2 \right] f_a(N_{\text{bin}})^2 + 2 \frac{N_{\text{bin}} - N_{\text{shift15}}}{N_{\text{bin}}} \frac{N_{\text{shift15}}}{N_{\text{bin}}} \frac{1}{N_{\text{bin}}} \sum_{i,j=1}^{N_{\text{bin}}} R(N_{\text{bin}} + j - i) \right\} \\
&= \frac{\text{Var}(b)}{N_{\text{bin}}} \left\{ \left[ \left( \frac{N_{\text{bin}} - N_{\text{shift15}}}{N_{\text{bin}}} \right)^2 + \left( \frac{N_{\text{shift15}}}{N_{\text{bin}}} \right)^2 \right] f_a(N_{\text{bin}})^2 \right. \\
&\qquad \left. + 2 \frac{N_{\text{bin}} - N_{\text{shift15}}}{N_{\text{bin}}} \frac{N_{\text{shift15}}}{N_{\text{bin}}} \left( \sum_{m=1}^{N_{\text{bin}}} \frac{m}{N_{\text{bin}}} R(m) + \sum_{m=1}^{N_{\text{bin}}-1} \frac{N_{\text{bin}} - m}{N_{\text{bin}}} R(N_{\text{bin}} + m) \right) \right\}.
\end{aligned}
\tag{A7}
$$

It follows that the correction function to apply on the standard deviation to take into account both the vertical partial correlation due to the electronic bandwidth and the redistribution in altitude registration is

$$
\begin{aligned}
f_{\text{corr}}(N_{\text{bin}}, N_{\text{shift15}}) = &\left\{ \left[ \left( \frac{N_{\text{bin}} - N_{\text{shift15}}}{N_{\text{bin}}} \right)^2 + \left( \frac{N_{\text{shift15}}}{N_{\text{bin}}} \right)^2 \right] f_a(N_{\text{bin}})^2 \right. \\
&\left. + 2 \frac{N_{\text{bin}} - N_{\text{shift15}}}{N_{\text{bin}}} \frac{N_{\text{shift15}}}{N_{\text{bin}}} \left( \sum_{m=1}^{N_{\text{bin}}} \frac{m}{N_{\text{bin}}} R(m) + \sum_{m=1}^{N_{\text{bin}}-1} \frac{N_{\text{bin}} - m}{N_{\text{bin}}} R(N_{\text{bin}} + m) \right) \right\}^{1/2}.
\end{aligned}
\tag{A8}
$$

## Appendix B: Surface detection

The aim of this procedure is to detect a surface echo in the near neighborhood region of the estimated surface altitude $\widehat{z_{surf}}$ given by a digital elevation model (DEM). The width of this region will vary according to surface type. Since we are highly confident of the surface altitude over the ocean (where $\widehat{z_{surf}} = 0$), we will only search in a very narrow range of altitudes for

profiles measured over the ocean. On the other hand, we are somewhat less confident of the DEM surface altitudes reported over land and even much less confident over Greenland and Antarctica, so our search regions over land will be larger. The surface echo can be very weak due to attenuation by aerosol and cloud layers above. Then, we try to detect even the weakest surface echo as long as it is substantially above background noise. This procedure is applied at single-shot resolution only. For each shot, the method is made up of the following steps:

1. Compute $\widehat{r_{\mathrm{surf}}}$, the estimated range of the surface, from $\widehat{z_{\mathrm{surf}}}$ and the satellite altitude $z_{\mathrm{sat}}$;

2. Compute $\Delta\beta'_b(\widehat{r_{\mathrm{surf}}})$, the standard deviation due to background noise in the $\beta'$ domain at the range $\widehat{r_{\mathrm{surf}}}$;

3. Compute $\widehat{i_{\mathrm{surf}}}$, the bin index of the estimated surface altitude, i.e., when $z(\widehat{i_{\mathrm{surf}}})$ is closest to $\widehat{z_{\mathrm{surf}}}$;

4. Define $\widehat{i_{\mathrm{surf}}} \pm \Delta i$, the range of the surface search region according to the International Geosphere/Biosphere Programme (IGBP) classification of the surface type at the lidar footprint:

   (a) If surface type is *Water* and $\widehat{z_{\mathrm{surf}}} = 0$, then $\Delta i = 2$ ($\equiv 60$ m),

   (b) Else if surface type is *Permanent-Snow*, then $\Delta i = 17$ ($\equiv 510$ m),

   (c) Else, $\Delta i = 5$ ($\equiv 150$ m);

5. Compute the derivatives of the lidar signal for bin index range 289–578 (8.2 to -0.5 km)

$$\left(\frac{\mathrm{d}\beta'}{\mathrm{d}z}\right)_i = \frac{\beta'_i - \beta'_{i-1}}{z_i - z_{i-1}};\tag{B1}$$

6. Determine $z_{\mathrm{min}}$ and $z_{\mathrm{max}}$, the altitudes of the minimum and maximum values of the derivatives in the surface search region, and $i_{\mathrm{min}}$ and $i_{\mathrm{max}}$, their respective bin index;

7. Determine $\beta'_{\mathrm{max}}$, the maximum signal magnitude lying between $z_{\mathrm{min}}$ and $z_{\mathrm{max}}$;

8. Sequentially test the three following rules to determine if we have identified a legitimate surface return:

   (a) $z_{\mathrm{min}} > z_{\mathrm{max}}$,

   (b) $i_{\mathrm{min}} - i_{\mathrm{max}} \leq N$ with $N = 2$ for the 532 nm channels and $N = 4$ for the 1064 nm channel,

   (c) $\beta'_{\mathrm{max}} > 3\Delta\beta'_b(\widehat{r_{\mathrm{surf}}})$;

9. If all rules passed, set surface bin index $i_{\mathrm{surf}}$ following these conditions:

   (a) if $\left(\frac{\mathrm{d}\beta'}{\mathrm{d}z}\right)_{i_{\mathrm{min}}-1} > 0$ **or** $\beta'_{i_{\mathrm{min}}-1} \leq 0$, then $i_{\mathrm{surf}} = i_{\mathrm{min}}$,

   (b) else (i.e., $\left(\frac{\mathrm{d}\beta'}{\mathrm{d}z}\right)_{i_{\mathrm{min}}-1} \leq 0$ **and** $\beta'_{i_{\mathrm{min}}-1} > 0$), $i_{\mathrm{surf}} = i_{\mathrm{min}-1}$ for the 532 nm channels and $i_{\mathrm{surf}} = i_{\mathrm{min}-2}$ for the 1064 nm channel.

10. If surface detection arose in a profile (profile horizontal index $p$) but not in the previous $(p-1)$ and the following $(p+1)$ profiles, then the surface detection is canceled if $i_{\mathrm{surf}} \notin \widehat{i_{\mathrm{surf}}} \pm 1$. This last step reduces the surface search region for isolated surface detection to prevent false detection in very attenuated region.

*Author contributions.* TVG led the design of 2D-McDA and wrote the manuscript. MV and DW provide the knowledge of the weaknesses of the previous detection algorithm and bring the main ideas to address them. ZL developed the correction function and contributed to discussion and feedback essential to the study.

*Competing interests.* The authors declare that they have no conflict of interest.

*Acknowledgements.* The authors are grateful to Brian Magill for helping improving the runtime of the algorithm and to Kenneth Beaumont and Brian Getzewich for running the algorithm on the cluster of the Atmospheric Science Data Center. Thibault Vaillant de Guélis' research was supported by an appointment to the NASA Postdoctoral Program at the NASA Langley Research Center, administered by Universities Space Research Association under contract with NASA.

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

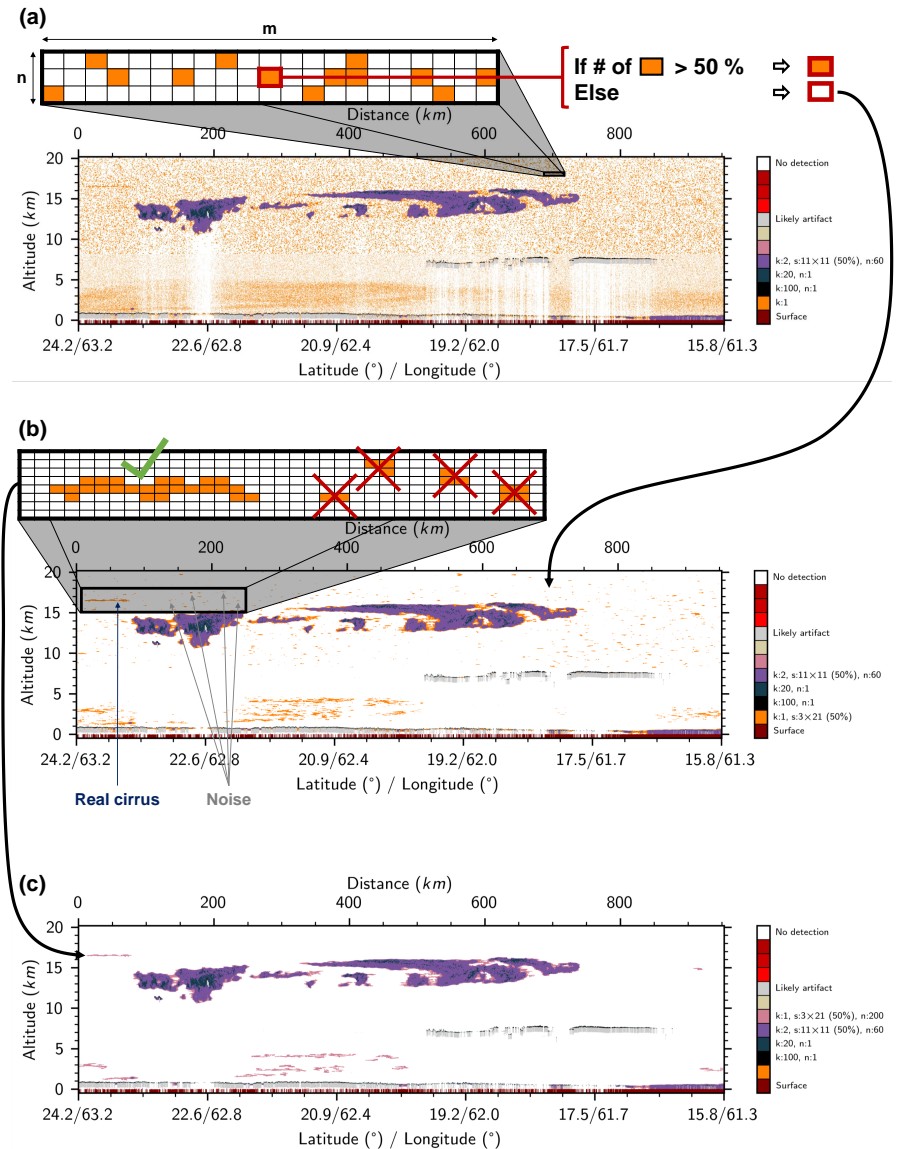

**Figure 6.** Illustration of the horizontal spatial coherence test window substep and the pattern size threshold rejection substep of 2D-McDA on the 532 nm parallel channel for the case study shown in Fig. 1. (a) Pixels which are over the detection threshold given by Eq. (6) with the value of $k = 1$ (orange). (b) Result after applying a 3×21 spatial coherence test window on the detected pixels. (c) Result after rejecting all patterns composed by less than $n = 200$ pixels (note that the insert on the top is just an illustration and does show the real content of the image portion). Before these substeps, surface is first detected (brown), then very strong signal ($k = 100$) occurring on highly reflecting liquid clouds is detected (black) and the 600 m below is flagged as "likely artifact" (gray) as it is the region where we see artifacts due to the time response of photomultiplier tubes (PMTs) in the 532 nm channels. Two detections were also made: one with $k = 20$ and another with $k = 2$, a 11×11 spatial coherence test window, and $n = 60$.

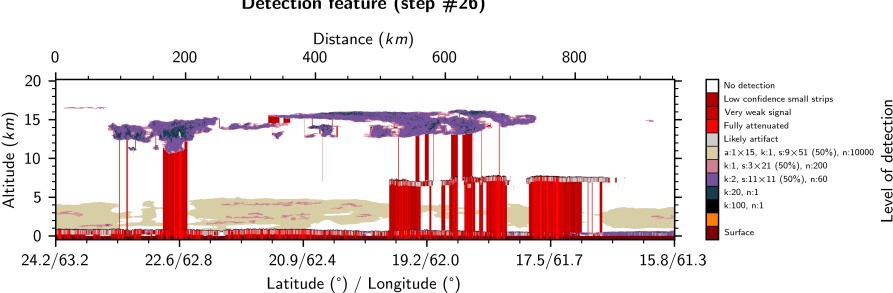

**Figure 7.** Final feature mask of the 532 nm parallel channel.

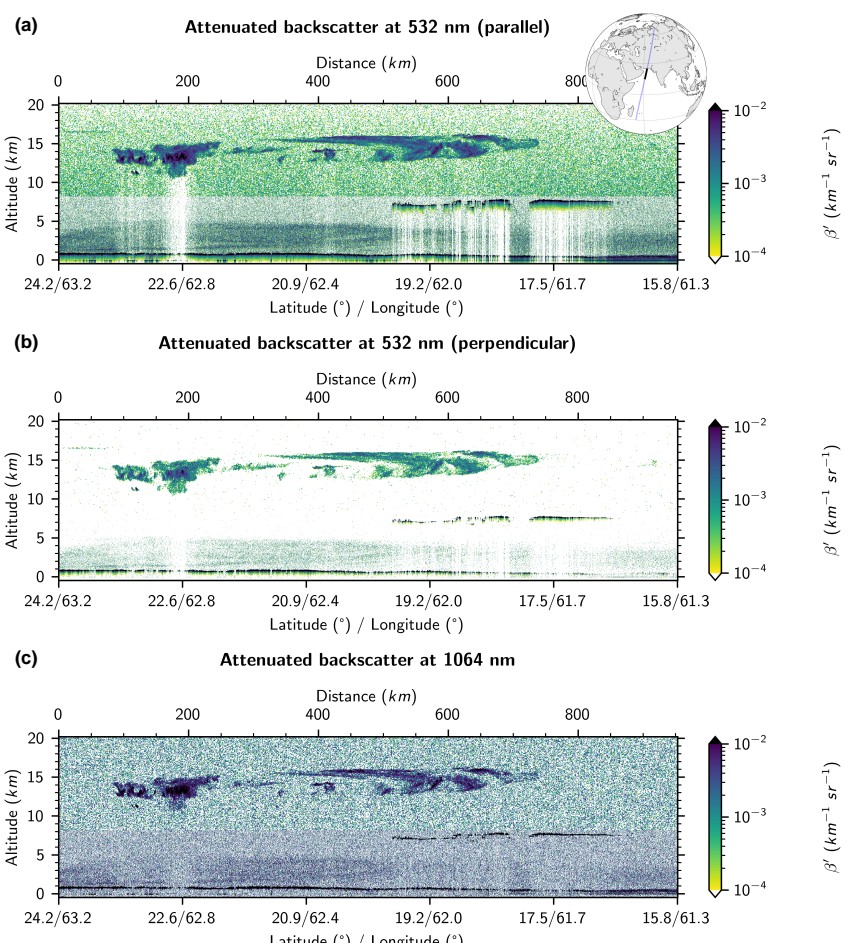

**Figure 8.** Curtain of lidar attenuated scattering ratio signal measured by CALIOP during nighttime observations on August 31, 21:46:37 UTC (start point), daytime observations: (a) at 532 nm parallel (same as Fig. 1), (b) at 532 nm perpendicular, and (c) 1064 nm.

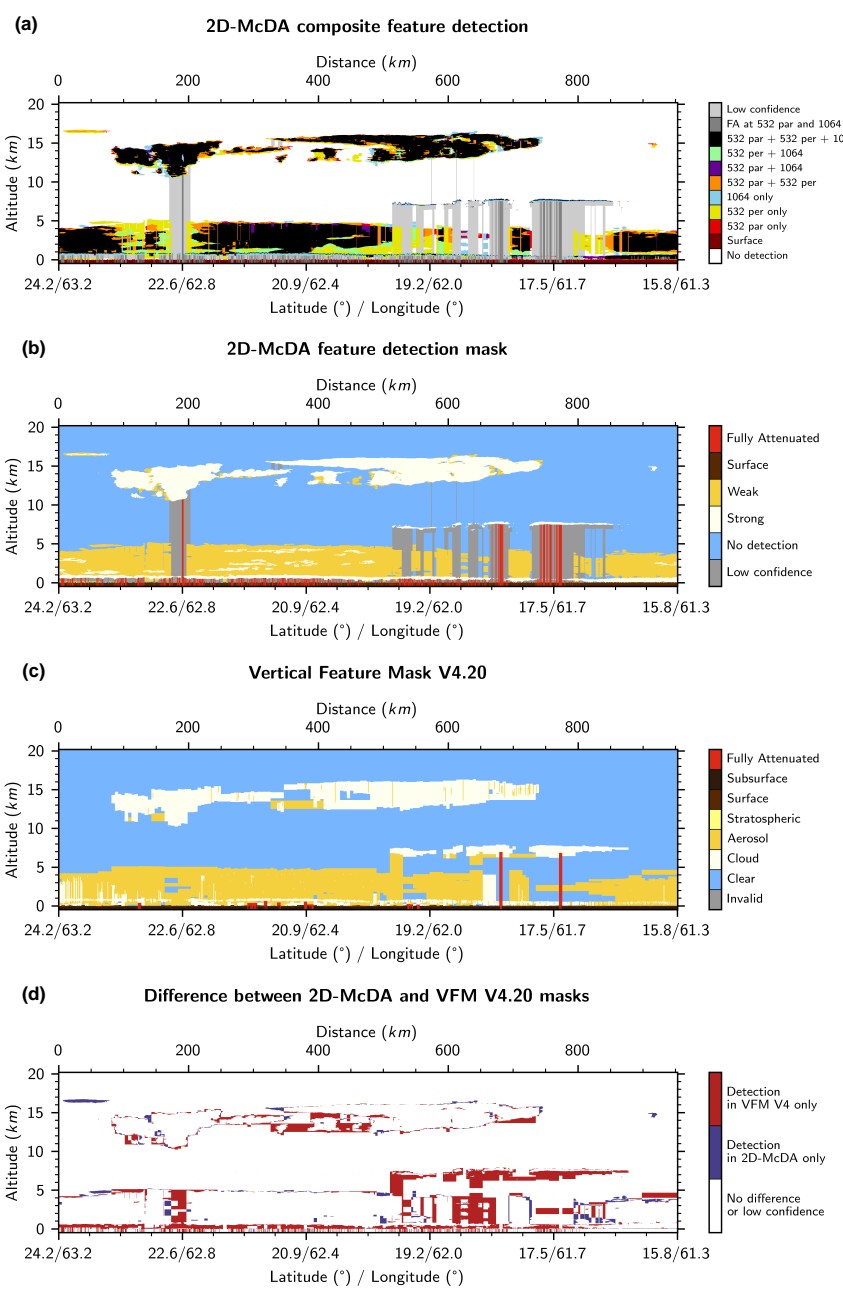

**Figure 9.** (a) Composite feature detection mask derived from signals shown in Fig. 8. (b) Same as (a) but using same colors than those used for VFM. "Strong" (white) are feature detected without averaging in at least one channel, others are flagged "Weak" (yellow). (c) VFM of the version 4 of the CALIOP data product. (d) Difference between new mask and VFM.

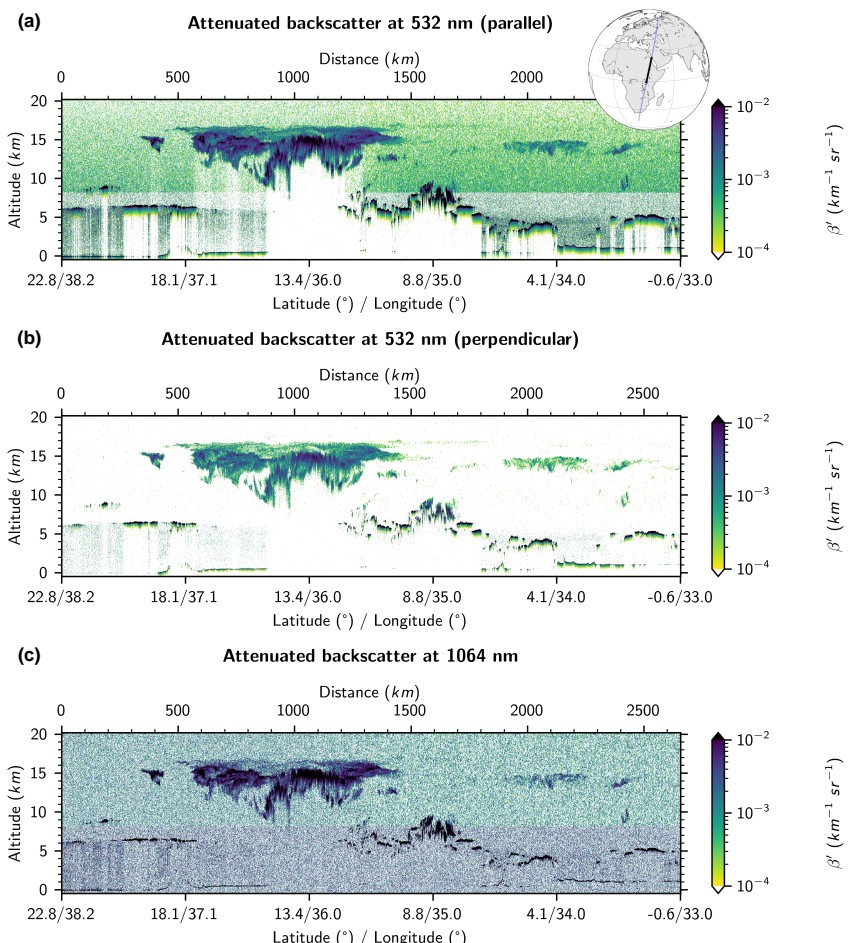

**Figure 10.** Curtain of lidar attenuated scattering ratio signal measured by CALIOP during nighttime observations on August 31, 2018, 23:25:54 UTC (start point), over Ethiopia: (a) at 532 nm parallel, (b) at 532 nm perpendicular, and (c) 1064 nm.

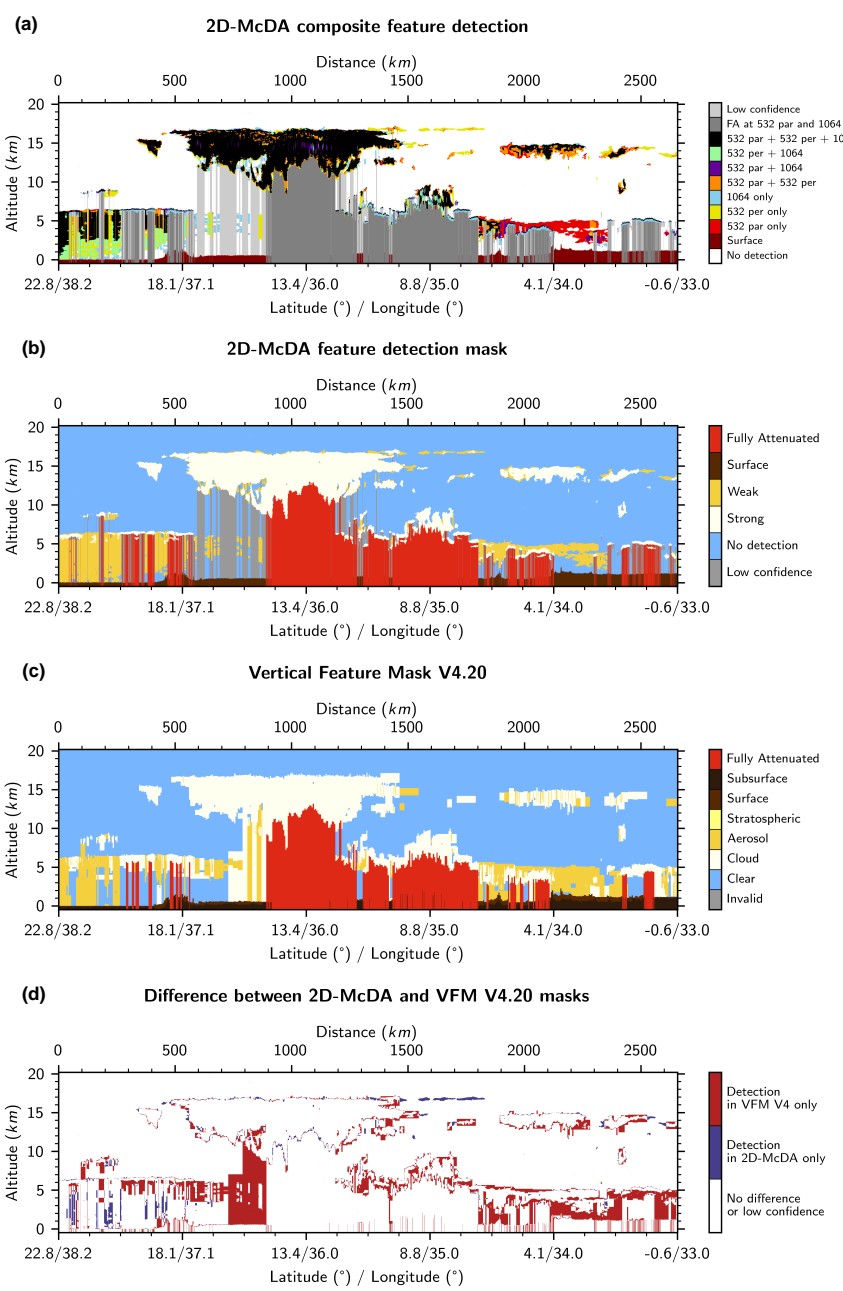

**Figure 11.** (a) Composite feature detection mask derived from signals shown in Fig. 10. (b) Same as (a) but using same colors than those used for VFM. "Strong" (white) are feature detected without averaging in at least one channel, others are flagged "Weak" (yellow). (c) VFM of the version 4 of the CALIOP data product. (d) Difference between new mask and VFM.

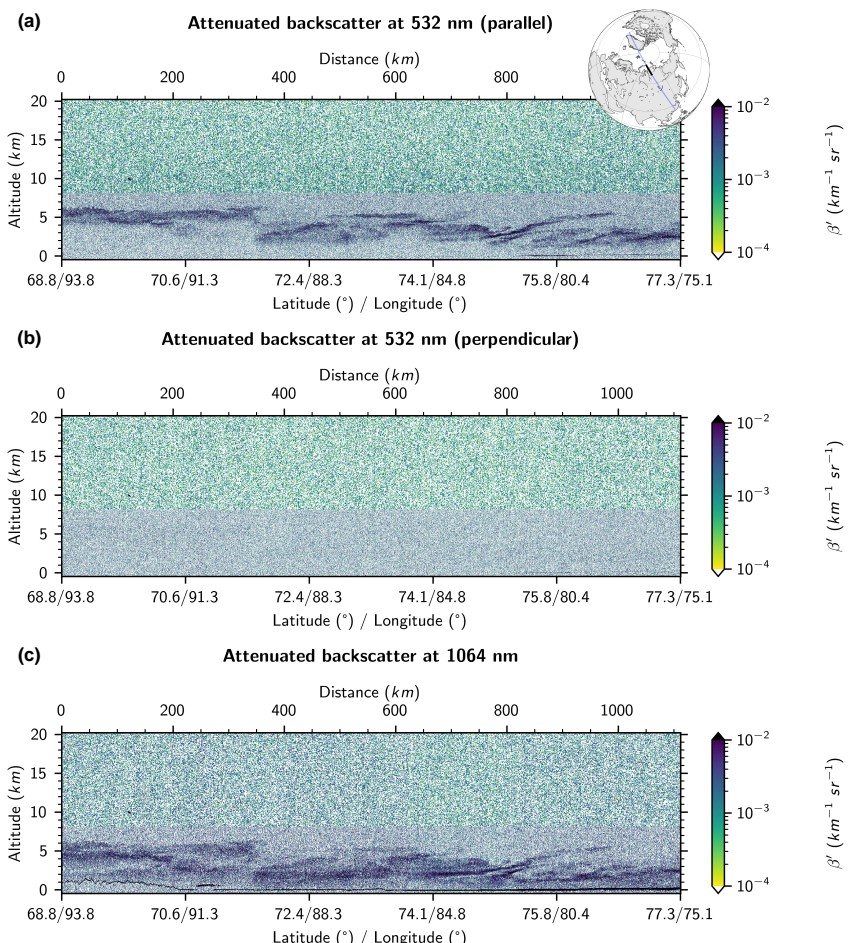

**Figure 12.** Curtain of lidar attenuated scattering ratio signal measured by CALIOP during a dense smoke event which occurred in Siberia on July 26, 2006, 06:00:25 UTC (start point), daytime observations: (a) at 532 nm parallel, (b) at 532 nm perpendicular, and (c) 1064 nm.

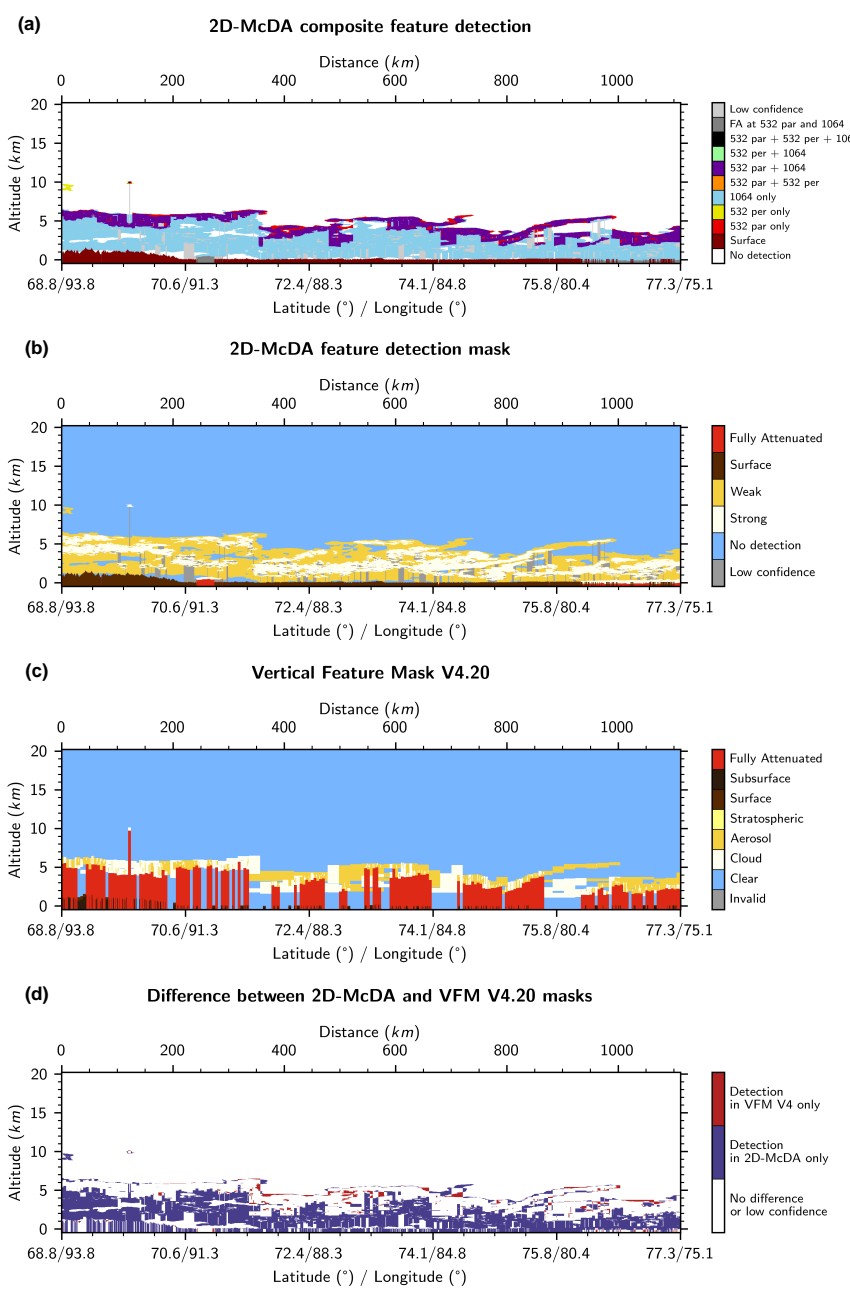

**Figure 13.** (a) Composite feature detection mask derived from signals shown in Fig. 12. (b) Same as (a) but using same colors than those used for VFM. "Strong" (white) are feature detected without averaging in at least one channel, others are flagged "Weak" (yellow). (c) VFM of the version 4 of the CALIOP data product. (d) Difference between new mask and VFM.

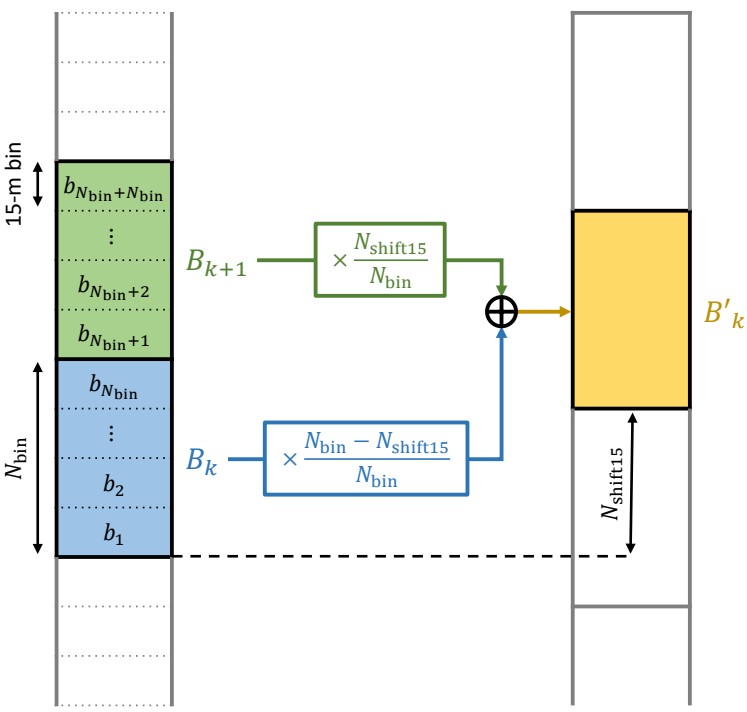

**Figure A1.** Scheme of redistribution in altitude registration.