# Peer review of "Two-dimensional and multi-channel feature detection algorithm for the CALIPSO lidar measurements"

_Atmospheric Measurement Techniques, 2020_

## Referee Comment (RC1) · Anonymous Referee #1 · 24 Nov 2020

This paper describes a new feature detection algorithm (2D-McDA) based on several channels and utilizing two dimensions (height vs. time/hor. distance) for the elastic lidar onboard the well-known CALIPSO platform.

The methodology is well explained and plots and table are used to illustrate this complex matter. Even though the paper is very technical, describing an algorithm for the spaceborne-lidar CALIOP, it is of very wide interest for the scientific community as CALIPSO products are widely used. Furthermore, I guess adapted versions of the algorithm could be also used for other lidars.

Some of the methodologies are based on empirically found thresholds, which is ok,

given the large data set and experience of the CALIPSO team. It is furthermore mostly well explained.

I have only some minor/specific comments which should be addressed before the paper can be published. I therefore compliment the authors on their excellent work and I am sure the new feature mask will drive new science based on CALIOP.

Specific comments:

Text:

Page 2. Line 42: Are this really profile processes? I guess you mean processing based on single profiles?

Page 2, line 43:"Scene processing" instead of "scene processes"

Page 3, line 45: Replace "If" by "Because" for better readability?

Page 3, line 47: improves–> improve

Page 3, bottom: Do you have any reference for the given formula? If not, more explanation is needed, because it is not evident why this formula can be used. Is there any theoretical background for this formula?

Page 4, line 75: Bins should be explained when used the first time. Even it might be clear for all lidar experienced persons it might not for others.

Page 6, 116 ff. Please make sure that you talk about the Earth' surface and not any other surface (like the one of clouds)

Page 7, caption Figure 7: Needs more explanation. Within the Figure caption it should be explained what k, s , n , a, d is and a proper reference to within the text should be given.

Page 7, line 150: why is data resolution duplicated? Is the image resolution always double? Or do you mean multiplied to. . .Please clarify!

Page 7: 154: What happens if you apply your e.g. 11x11 filter at the edges, e.g. close to the surface, do you have to decrease the window extent accordingly? Or is the number of pixel reduced due to the averaging?

Page 10, 178: I would make it more clear: small clusters of pixels→ small clusters of noise pixels.

Page 10, 179-180: This step is not clear to me. what is the minimum numeric threshold in this case?

Page 10: 193-195: I really do not understand this sentence: "Pixels flagged as AFA are those vertically between two detected features if more than 30 % of them have their signal less than 10 % of the threshold for the 532 nm parallel channel, or more than 90 % and 85 % of them have their signal less than the threshold for the 532 nm perpendicular and 1064 nm channel respectively. " Please rephrase!

Page 10, line 202-203: In table 1 there is no alpha only a! Can you describe more what an edge-preserving Gaussian sliding window is or give reference?

Page 10, line 207: What does "unique feature" mean? Is the low confidence band counted or just detected as single feature?

Page 11: 228: . . . layer is transparent . . .: Is transparent the correct word? Would semi-transparent be better? Please emphasize that attenuation is also taking place at 1064 but less than at 532.

Page 11: I propose to discuss case 4.2. first. Because you explain much more for this case what can be seen in your provided plots. Thus Case 4.1 should come afterwards or more explanation need to be added to case 4.1.

Conclusion:

250-252: This sentence is very complicated, please try to rephrase.

271-274: Concerning the 1064 apparent cloud base: Do you expect no multiple scattering effects which influence that altitude? Is thus a direct link to the amount of long-wave radiation escaping the Earth at the top of the atmosphere as stated in the outlook reasonable?

275-277: The statement that the 1064 channel can generally penetrate dense smoke layers is for sure not true. It should be rather stated that due to the weaker attenuation, the 1064 channels can more often penetrate the full vertical extent compared to the 532 nm channel.

280: For aerosol-cloud interactions you do not necessarily need a more accurate optical depth, instead you need the vertical profile (i.e. the extinction or better the concentration at the height were the cloud is forming) . . ... Please rephrase.

283-287: I do not understand why for "taking the full advantage of the new algorithm", companion scene classification algorithm need to be developed. Why can't you simply use the one you just presented?

Appendix:

In general, the Appendix seems to need a little bit more explanation, e.g.:

Page 14, 305ff: "In general, shift for a few range bins is needed for the full resolution (30 m) samples. Redistribution and rebinning of two neighboring samples whose range resolutions are coarse (60 m, 180 m, and 300 m) are performed. . ..." What are a few range bins? How is the redistribution done? Maybe introduce Fig. A1 first and explain the methodology briefly or give reference.

Page 16. Line 341: step 7: why is the $z_{max}$ squared? Can you give explanation?

Page 16, line 350: I do not understand step 10 as in my opinion it is in contradiction to step 4. Why is only $i_{surf}$+-1 allowed? Can you explain?

Furthermore, please briefly introduce indices "i" and "k" and explain that "i" is used for the vertical while "k" is used for the horizontal dimension.

Code availability:

This is usually a section now included in Copernicus paper: Do you plan to make your code available?

---

## Referee Comment (RC2) · Anonymous Referee #2 · 6 Dec 2020

In this paper a 2D feature algorithm (2D-McDA) based on the two 532 and 1064nm channels from the CALIOP lidar onboard the CALIPSO satellite. This work clearly shows the advantages of using horizontal and vertical coherence between pixels for detection of features. The 2D-McDA uses approaches taken from the field of image processing algorithm that examines full resolution lidar signals and can therefore identify finer details than can be retrieved by more standard multi-resolution averaging schemes. The use of the depol. channel and 1064nm channel introduce many new possibilities, like the detection of smoke and thin cirrus as described in the paper.

The paper is well written and easy to follow. The algorithm and all its details are well

explained and shown in corresponding images for each of the individual steps.Looking at the successful implementation, this work will very likely lead to the use of similar approaches in the future for similar type of signals.

There are a few minor comments which have to be addressed before publication.

- Figure 2: Caption, please mention here that the jumps in the red line shows the change in vertical resolution. This is mentioned in the text (line 92-93), but it would help people browsing through the paper.

- Page 5, line 98: 'For practical reasons, the layer detection scheme uses ..'. Explain practical reasons: most likely what you mean is that the dynamical range of the signal is smaller and linear instead of the exponential backscatter range.

- Page 6, line 130: Remove 'refer to'

- Page 9, line 172-174: 'This detection procedure ... very extended ones.' Are we talking here about the three options per channel in the table? Please describe this more specific.

- Page 10, line 202: 'We then average the remaining signal (here the attenuated scattering ratios) using an edge-preserving Gaussian sliding window that extends over 5 km (15 profiles) horizontally and a single range bin vertically (a in Table 1). 'Have you filled in the area's which had features with noise or are you assuming 0 in those area's? In the latter case you may loose connections between features due to adding no signals.

- Page 10, line 205: 'faint layers is mainly in the horizontal direction'. Please elaborate! Can you specify this more, i.e. aerosol and thin cirrus regions have a long horizontal scale, for aerosols this can go up to 50-100 km. Also the horizontal window allows for the detection of thin layers close to each other vertically

- Page 12 line 241: 'yellow and white colors do not discriminate aerosol from cloud, as in the VFM, but instead simply differentiate weak from strong features based on
whether the feature detection required data averaging (yellow) or not (white).' This is actually a very important feature of these kind of masks and it is mentioned here for the first time explicitly. You need to mention this separation on strong and weak instead of cloud & aerosol in the abstract, introduction and conclusion as it is conceptually very different from the standard methods that you only look for non-molecular and not the type.

- Page 13 line 283-286 : 'While .... algorithms'. You are very correct with the future outlook that you would like a 2D feature typing in the end, however there are lower hanging fruits to pick first. The detected features will improve and guide the smoothing strategies of the data. This will already show large improvements in the retrievals, i.e. strong and weak features are separately smoothed to calculate the local profile and provide you with the best possible lidar, color and depol. ratio's to enable a better typing and aerosol sub-typing. I think a somewhat broader discussion on how your product can be used is needed here.

―――――――――――――――――――――

---

## Author Comment (AC1)

Point-by-point responses to the reviews

We thank the reviewers for their reviews and valuable comments. The manuscript has been modified according to the reviewers' suggestions. Below we offer specific, item-by-item response to all the reviewers' comments.
* * *
**Anonymous Referee #1**

This paper describes a new feature detection algorithm (2D-McDA) based on several channels and utilizing two dimensions (height vs. time/hor. distance) for the elastic lidar onboard the well-known CALIPSO platform.

The methodology is well explained and plots and table are used to illustrate this complex matter. Even though the paper is very technical, describing an algorithm for the spaceborne-lidar CALIOP, it is of very wide interest for the scientific community as CALIPSO products are widely used. Furthermore, I guess adapted versions of the algorithm could be also used for other lidars.

- **Response:** Yes, we certainly envision adapting our algorithm for use by other lidars. However, because we have not yet done so, we refrain from commenting on wider applicability in the manuscript.

Some of the methodologies are based on empirically found thresholds, which is ok, given the large data set and experience of the CALIPSO team. It is furthermore mostly well explained.

I have only some minor/specific comments which should be addressed before the paper can be published. I therefore compliment the authors on their excellent work and I am sure the new feature mask will drive new science based on CALIOP.

Specific comments:

Text:

Page 2. Line 42: Are this really profile processes? I guess you mean processing based on single profiles?

Page 2, line 43:"Scene processing" instead of "scene processes"

- **Manuscript modification:**

  ➤ In Introduction (l. 42–43): "Because the layer detection algorithm are applied successively to individual  1D profiles (either single shot or averaged), we define them collectively as *profile-based processes*. We also define a second, more comprehensive class of methods  as *scene processes*. Scene processes can take advantage of [...]".

Page 3, line 45: Replace "If" by "Because" for better readability?

Page 3, line 47: improves–> improve

- **Manuscript modification:**

  ➤ In Introduction (l. 45–47): "While edge detection techniques based on 2D gradient search routines are not well-suited for spatial analysis of lidar data (Vaughan et al., 2005), methods based on sliding window operations have been shown to greatly improve the feature shape detection [...]".

Page 3, bottom: Do you have any reference for the given formula? If not, more explanation is needed, because it is not evident why this formula can be used. Is there any theoretical background for this formula?

- **Manuscript modification:**

  ➤ In Sect. 2 (l. 68–72): "A feature, i.e., a cloud or an aerosol layer, appears as an extended and contiguous region of enhanced attenuated backscatter signal that rises significantly above the expected clear sky (molecules only) value.

atmospheric feature.However, not all signals that exceed the expected values of $\widehat{\beta'_m(r)}$ necessarily indicate the presence of  features; instead such excursions are often caused by noise. To distinguish features from the ambient (but noisy) clear sky signals, a first step is to determine a threshold above which signals can be confidently attributed to enhanced scattering arising from clouds or aerosols. We construct this threshold by first calculating the expected molecular attenuated backscatter, $\widehat{\beta'_m(r)}$, to which we add $k$ times the expected noise-induced standard deviation of the molecular signal. The resulting range-dependent threshold is the sum of $\widehat{\beta'_m(r)}$ and, based on error propagation theory (e.g., Bevington and Robinson, 2003), $k$ times the root mean square (RMS) of the standard deviations due to both range-independent and range-dependent noise sources.".

➢ In Sect.2 (l. 80–81): "RMS".

Page 4, line 75: Bins should be explained when used the first time. Even it might be clear for all lidar experienced persons it might not for others.

▪ **Response:** We agree with the reviewer that more details should be provided for non-lidar experienced persons. Many thanks.

▪ **Manuscript modification:**

➢ In Sect. 2 (l. 72–75): "In constructing thresholds to be applied to CALIOP data, one must take into account the onboard signal averaging that is applied to the backscatter measurements. Because the CALIPSO satellite has limited telemetry bandwidth, the backscatter data is averaged both vertically and horizontally before the data is downlinked from the satellite, with increasing amounts of averaging applied to data acquired at higher altitudes (Hunt et al., 2009). As an example, signals acquired between 8.2 km and 20.2 km are averaged horizontally over three consecutive lidar pulses and vertically for four full resolution (15 m) range bins. Consequently, the downlinked profile data from within this region have been averaged over 12 full resolution onboard range bins. We compute a range-dependent threshold specifically tailored for the CALIOP profiles using
[Eq. (3)]
where $\Delta\beta'_b(r)$ is the standard deviation due to background noise (range-independent[1]), $\Delta\widehat{\beta'_m(r)}$ is the expected standard deviation due to the shot noise (range-dependent) in the expected clear sky, and $N(r)$ is the number of bins averaged onboard.".

Page 6, 116 ff. Please make sure that you talk about the Earth' surface and not any other surface (like the one of clouds)

▪ **Manuscript modification:**

➢ In Sect. 3.1 (l. 110–111): "[...] we perform first an independent detection of the Earth's surface .".

➢ In Sect. 3.1 (l. 116): "The Earth surface detection algorithm [...]".

Page 7, caption Figure 7: Needs more explanation. Within the Figure caption it should be explained what k, s , n , a, d is and a proper reference to within the text should be given.

▪ **Response:** Thank you for noting these were missing.

▪ **Manuscript modification:**

➢ In Fig. 4 caption (l. 138): "Flowchart of the two-dimensional and multi-channel feature detection algorithm (2D-McDA). $d$ is the detection step, $k$ is the number of total noise standard deviations used in the detection threshold Eq. (3), $s$ is the size of the window used for the spatial coherence test, $n$ is the minimum number of pixels in each pattern, and $a$ is the size of the averaging window. See the algorithm description in Sect. 3 and the coefficient values in Table 1.".

Page 7, line 150: why is data resolution duplicated? Is the image resolution always double? Or do you mean multiplied to. . .Please clarify!

- **Manuscript modification:**

  - In Sect. 3.2.1 (l. 149–150): "Scattering ratios in regions where the data resolution is coarser than the image resolution (30 m vertically × 1/3 km horizontally) are duplicated as necessary to match the image resolution. For example, between 8.2 and 20.2 km, the spatial resolution of the signal is 1 km horizontally × 60 m vertically. These values are replicated 12 times to populate the corresponding area in the 30 m × 1/3 km scattering ratio image.".

Page 7: 154: What happens if you apply your e.g. 11x11 filter at the edges, e.g. close to the surface, do you have to decrease the window extent accordingly? Or is the number of pixel reduced due to the averaging?

- **Response:** Many thanks for your comment. At the edges, the number of pixels considered in the window is reduced accordingly to the number of pixels of the window outside the image. Please note that, because the CALIOP data is horizontally continuous, this normally only occurs at the top and bottom of the image and at the day-night terminators.

- **Manuscript modification:**

  - In Sect. 3.2.1 (l. 163–165): "Other flagged pixels (i.e., "Surface", detection $\leq d - 2$, "Likely artifact", "Fully attenuated", "Almost fully attenuated", and "Low confidence small strips" (to be described in detail later in this subsection and subsection 3.2.2) and pixels outside the window when at the top or bottom edges of the image are not considered in the window and the total number of candidate pixels in the window is decreased accordingly.".

Page 10, 178: I would make it more clear: small clusters of pixels→ small clusters of noise pixels.

- **Manuscript modification:**

  - In Sect. 3.2.1 (l. 178–179): "However, many small clusters of noise pixels persist.".

Page 10, 179-180: This step is not clear to me. what is the minimum numeric threshold in this case?

- **Response:** Each time we apply a spatial coherence test window, which smooths the shape of detected pattern and remove isolated noisy detected pixels, the next step is to remove small clusters of pixels by applying a minimum numeric threshold of connected pixel $n$.

- **Manuscript modification:**

  - In Sect. 3.2.1 (l. 179–180): "By applying the minimum numeric threshold of connected pixel $n$ on the detected pattern, we are able to remove small cluster due to noise while keeping the real cirrus (Fig. 6c).".

Page 10: 193-195: I really do not understand this sentence: "Pixels flagged as AFA are those vertically between two detected features if more than 30 % of them have their signal less than 10 % of the threshold for the 532 nm parallel channel, or more than 90 % and 85 % of them have their signal less than the threshold for the 532 nm perpendicular and 1064 nm channel respectively. " Please rephrase!

- **Response:** We agree with the reviewer that this sentence needs to be rephrased. Thank you.

- **Manuscript modification:**

  - In Sect. 3.2.2 (l. 192–198): "Second, the contiguous pixels located in the vertical extent between two detected features are flagged as "Almost fully attenuated" (AFA) whenever the backscatter intensity falls below an empirically determined threshold. For the 532 nm parallel channel, these pixels are flagged as AFA

between two detected features if more thanm population hasbackscatter intensities that are less than 10 % of the corresponding detection thresholds. These pixels are flagged AFA in the 532 nm perpendicular channel whenever the signals in more than 90 % m population fall below (100 % of) the corresponding threshold values.  To be flagged AFA in the 1064 nm channel,  more than 85 % of the population must have signal less than (100 %) of the corresponding threshold. In all cases, these AFA thresholds were determined experimentally and are tunable. Finally, the horizontal distance between successive (A)FA columns can be very small and the likelihood of confidently detecting features in these narrow gaps is very low. For this reason, the data in all horizontal  extents smaller than 5 km (15 profiles) that lie between (A)FA columns  are flagged as "Low confidence small strips".".

Page 10, line 202: In table 1 there is no alpha only a!

- **Response:** It is actually the character "a" which is on this line. The default font of LaTeX equation is used here and appears to be different from the font used Table 1 and Fig. 4.

- **Manuscript modification:** The font of parameters in Table 1 and Fig. 4 has been changed in order to match the font used in the text.

Page 10, line 202-203: Can you describe more what an edge-preserving Gaussian sliding window is or give reference?

- **Response:** We use a simple Gaussian sliding window which, associated with a spatial coherence test, preserves edges.

- **Manuscript modification:**

  ➢ In Sect. 3.2.3 (l. 202–203): "We then average the remaining signal (here the attenuated scattering ratios) using a Gaussian sliding window that extends over 5 km (15 profiles) horizontally and a single range bin vertically (a in Table 1). Using a sliding window, instead of the fixed window used in the CALIOP feature detection algorithm, provides much improved resolution of the horizontal edges position of faint features (1/3 km instead of 5, 20, or 80 km) and makes it possible to detect non-uniform horizontal edges.".

Page 10, line 207: What does "unique feature" mean? Is the low confidence band counted or just detected as single feature?

- **Manuscript modification:**

  ➢ In Sect. 3.2.3 (l. 206–208): "Note too that  horizontally adjacent features  separated only by a low confidence vertical band (i.e., pixels classified as FA, AFA, and/or small strips)  are considered as a  single, merged feature when counting the number of connected pixels. Some examples of this horizontal merging are seen in the smaller fragments of the aerosol layer found at about 4 km and an along track distance of 500 km to 750 km in Fig. 7.".

Page 11: 228: . . . layer is transparent . . .: Is transparent the correct word? Would semi-transparent be better? Please emphasize that attenuation is also taking place at 1064 but less than at 532.

- **Response:** Yes, "semi-transparent" is better. Many thanks.

- **Manuscript modification:**

  ➢ In Sect. 4.1 (l. 228): "However, at 1064 nm the dense smoke layer is semi-transparent because the 1064 nm signals are attenuated significantly less than at 532 nm. Then,  the surface is readily detected at 1064 nm (Fig. 10c).".

Page 11: I propose to discuss case 4.2. first. Because you explain much more for this case what can be seen in your provided plots. Thus Case 4.1 should come afterwards or more explanation need to be added to case 4.1.

- **Manuscript modification:** Sect. 4.1 and 4.2 have been inverted.

Conclusion:

250-252: This sentence is very complicated, please try to rephrase.

- **Manuscript modification:**

  ➢ In Conclusions (l. 250–252): "The cloud and aerosol layer detection boundaries algorithms used to generate reported in the standard CALIPSO data products retrieve cloud and aerosol layer boundariesare detected by scanning sequences of 532 nm attenuated scattering ratio individual lidar profiles constructed at varying increasingly coarser horizontal averaging resolutions.".

271-274: Concerning the 1064 apparent cloud base: Do you expect no multiple scattering effects which influence that altitude? Is thus a direct link to the amount of longwave radiation escaping the Earth at the top of the atmosphere as stated in the outlook reasonable?

- **Response:** Multiple scattering effects do indeed influence the level of complete attenuation of the lidar signal (i.e. the apparent cloud base altitude) and is taking into account when retrieving this level. This topic is discussed explicitly in Vaillant de Guélis et al. (2017a), section 6.5.

  – Vaillant de Guélis, T., Chepfer, H., Noel, V., Guzman, R., Dubuisson, P., Winker, D. M., and Kato, S.: The link between outgoing longwave radiation and the altitude at which a spaceborne lidar beam is fully attenuated, Atmos. Meas. Tech., 10, 4659–4685, https://doi.org/10.5194/amt-10-4659-2017, 2017a.

- **Manuscript modification:**

  ➢ In Conclusions (l. 271–272): "Theis apparent cloud base altitude, level which results from both attenuation of the direct beam and multiple scattering effects, has been directly linked with the amount of longwave radiation escaping the Earth at the top of the atmosphere [...]".

275-277: The statement that the 1064 channel can generally penetrate dense smoke layers is for sure not true. It should be rather stated that due to the weaker attenuation, the 1064 channels can more often penetrate the full vertical extent compared to the 532 nm channel.

- **Manuscript modification:**

  ➢ In Conclusions (l. 275–277): "Within smokes, the 1064 nm signals are attenuated significantly less than at 532 nm and hence can generally more often penetrate the full vertical extent of these layers.".

280: For aerosol-cloud interactions you do not necessarily need a more accurate optical depth, instead you need the vertical profile (i.e. the extinction or better the concentration at the height were the cloud is forming) . . ... Please rephrase.

- **Response:** We agree with the reviewer. Thank you.

- **Manuscript modification:**

  ➢ In Conclusions (l. 279–281): "The full detection of those layers will lead to more accurate aerosol optical depth retrievals, which will improve estimates of the radiation budget and interactions with clouds. Profiling the full depth of the smoke layer will also help to understand whether the layer is in contact with underlying clouds and able to affect cloud microphysics.".

283-287: I do not understand why for "taking the full advantage of the new algorithm", companion scene classification algorithm need to be developed. Why can't you simply use the one you just presented?

- **Response:** The algorithm presented in this paper is a feature detection algorithm. Whether these features are clouds or aerosols is yet to be determined, as is the thermodynamic phase of any clouds that are detected, and the subtype of any aerosols detected. Accomplishing these essential tasks requires the development of new set of 2D scene process that, like the 2D-McDA, will take advantage of spatial correlations more accurately classify all detected features.

- **Manuscript modification:**

  ➤ In Conclusions (l. 283–286): "However, while fundamentally important, feature detection is only the first step in extracting comprehensive suite of geophysical parameters from raw lidar measurements. tTaking full advantage of the improved spatial analyses delivered by the 2D-McDA thus requires the development of a companion set of 2D scene processes to replace the 1D profile-based processes that are currently used in the CALIPSO retrieval architecture to perform the essential tasks of discriminating between clouds and aerosols, identifying cloud thermodynamic phase, and classifying aerosols by type.".

Appendix:

In general, the Appendix seems to need a little bit more explanation, e.g.:

Page 14, 305ff: "In general, shift for a few range bins is needed for the full resolution (30 m) samples. Redistribution and rebinning of two neighboring samples whose range resolutions are coarse (60 m, 180 m, and 300 m) are performed. . .." What are a few range bins? How is the redistribution done? Maybe introduce Fig. A1 first and explain the methodology briefly or give reference.

- **Response:** Thank you for your comment. We've added an introduction paragraph to Appendix A1 and made a few changes as noted below.

- **Manuscript modification:**

  ➤ In Appendix A1 (l. 291): "A correction should be applied to Eq. (3) due to the fact that the nominal sample range interval (15 m) of the lidar is smaller than its range resolution (~40 m) determined by the electronic bandwidth (2 MHz; Hunt et al., 2009). Consequently, a 15-m sample bin is partially correlated with the two bins above and the two bins below. As a result, vertical averaging of several 15-m bins $N_{\text{bin}}$ does not reduce the noise standard deviation as much as it would if the samples were independent. A function $f_a(N_{\text{bin}}) > 1$ is then applied to correct from this partial correlation. This function is evaluated as follows.".

  ➤ In Appendix A1 (l. 299–300): "It follows that the correction function to apply on the total noise standard deviation in Eq. (3) to take into account the vertical partial correlation due to the electronic bandwidth is [...]".

  ➤ In Appendix A2 (l. 303–309): "An additional correlation arises from the data redistribution in the altitude registration of level 0 data during the level 1A processing. Indeed, the altitude of the sample bins of a raw data profile are recalculated with more accurate information about the satellite altitude and laser viewing angle in the data processing on ground. A shift for a few range bins (no more than three in most of the cases) can be needed for the full resolution (30 m) samples.  The number of 30-m bins shifted $N_{\text{shift30}}$ (which we express below in terms of an equivalent number of 15-m bins shifted $N_{\text{shift15}}$) only add correlation to regions in the profile data where the vertical range resolution is coarser than 30 m, i.e. where the vertical range resolution is 60 m, 180 m, and 300 m (Winker et al, 2006). Indeed, in those regions, a vertical shift by $N_{\text{shift30}}$ 30-m bins lead to the necessity of rebinning two neighboring bins larger than 30 m which introduce additional correlation to those bins. When there is a shift of $N_{\text{shift15}}$ 15-m bins (an even number since shifts are performed at 30 m resolution), each new shifted bin $B'_k$, with vertical resolution coarser than 30 m, is computed from the weighted average of the two original bins (with $N_{\text{bin}}$ size resolution) it steps across $B_k$ and $B_{k+1}$ (Fig. A1) following [...]".

Page 16. Line 341: step 7: why is the z_max squared? Can you give explanation?

- **Response:** $z_{\max}$ is not squared. "2" is here a footnote reference. As this footnote does not bring any fundamental information, it has been removed to avoid any confusion.

Page 16, line 350: I do not understand step 10 as in my opinion it is in contradiction to step 4. Why is only i_surf+-1 allowed? Can you explain?

Furthermore, please briefly introduce indices "i" and "k" and explain that "i" is used for the vertical while "k" is used for the horizontal dimension.

- **Response:** Step 10 means that $\Delta i$ defined in step 4 is reduced to $\Delta i = 1$ when surface detection arises in an isolated profile (i.e. no surface detection in the previous and following profiles). This additional step prevents false surface detection in very attenuated area.

  We have changed horizontal index "k" by "p" in order to avoid confusion with the parameter "k" used in the paper.

- **Manuscript modification:**

  - In Appendix B (l. 350–351): "10. If surface detection  arose in a profile  (profile horizontal index $p$) but not in the previous ($p-1$) and the following ($p+1$) profiles, then the surface detection is canceled if $i_{\mathrm{surf}} \notin \overline{i_{\mathrm{surf}}} \pm 1$ . This last step reduces the surface search region for isolated surface detection to prevent false detection in very attenuated region.".

Code availability:

This is usually a section now included in Copernicus paper: Do you plan to make your code available?

- **Response:** Yes, we would like to eventually release the code. However, because the first author was a 'US government contractor' (i.e., employed by entities under contract to the United States government) during the development of the code, we cannot distribute the code until we have clear legal authorization to do so from all necessary authorities.
* * *
**Anonymous Referee #2**

In this paper a 2D feature algorithm (2D-McDA) based on the two 532 and 1064nm channels from the CALIOP lidar onboard the CALIPSO satellite. This work clearly shows the advantages of using horizontal and vertical coherence between pixels for detection of features. The 2D-McDA uses approaches taken from the field of image processing algorithm that examines full resolution lidar signals and can therefore identify finer details than can be retrieved by more standard multi-resolution averaging schemes. The use of the depol. channel and 1064nm channel introduce many new possibilities, like the detection of smoke and thin cirrus as described in the paper.

The paper is well written and easy to follow. The algorithm and all its details are well explained and shown in corresponding images for each of the individual steps. Looking at the successful implementation, this work will very likely lead to the use of similar approaches in the future for similar type of signals.

There are a few minor comments which have to be addressed before publication.

- Figure 2: Caption, please mention here that the jumps in the red line shows the change in vertical resolution. This is mentioned in the text (line 92-93), but it would help people browsing through the paper.

- ▪ **Response:** Thank you for your comment.

- ▪ **Manuscript modification:**

  - ➢ In Fig. 2 caption (l. 73): "[…] Jumps in the lidar signal and threshold at –0.5 km, 8.2 km, and 20.2 km reveal the change of onboard averaging resolution.".

  - ➢ In Sect. 2 (l. 93): "Jumps at -0.5 km, 8.2 km, and 20.2 km reveal the change of onboard averaging resolution.".

- Page 5, line 98: 'For practical reasons, the layer detection scheme uses ..'. Explain practical reasons: most likely what you mean is that the dynamical range of the signal is smaller and linear instead of the exponential backscatter range.

- ▪ **Manuscript modification:**

  - ➢ In Sect. 2 (l. 98): "Like the current CALIOP layer detection  algorithm, the 2D-McDA is applied to profiles of attenuated scattering ratios, defined as
    [Eq. (5)]
    .".

- Page 6, line 130: Remove 'refer to'

- ▪ **Response:** Done as suggested.

- Page 9, line 172-174: 'This detection procedure ... very extended ones.' Are we talking here about the three options per channel in the table? Please describe this more specific.

- ▪ **Manuscript modification:**

  - ➢ In Sect. 3.2.1 (l. 172–174): "This detection procedure is applied several times (the successive detection level $d$ of Table 1) with different thresholds, different spatial coherence test window $s$, and different limits on the number of connected pixels required $n$ (Table 1) in order to detect all layers from the most evident, very strong patterns to the very faint ones, and from geometrically small patterns to very extended ones.".

  - ➢ In Table 1 caption (l. 163): "[...] at each detection  level $d$.".

- Page 10, line 202: 'We then average the remaining signal (here the attenuated scattering ratios) using an edge-preserving Gaussian sliding window that extends over 5 km (15 profiles) horizontally and a single range bin vertically

(a in Table 1). 'Have you filled in the area's which had features with noise or are you assuming 0 in those area's? In the latter case you may loose connections between features due to adding no signals.

- **Response:** Many thanks for your comment. Features detected at a lower detection level are flagged as detected and the corresponding signal is masked in the scattering ratio image. If such pixels are in the averaging window, they are ignored and the number of points in the window is decreased accordingly. If the center pixel of the averaging window (i.e. the pixel to which the averaging is applied) is a low confidence pixel (i.e. "Likely artifact", "Fully attenuated", "Almost fully attenuated", or "Low confidence small strips"), then the averaging window is indeed applied (unless all pixels of the window are already flagged) and corresponding pixel in the feature detection mask is "unflagged" until the end of the detection level processing, after which its low confidence flag is put back. This allows us to maintain connections between features separated by a few low confidence pixels.

- **Manuscript modification:**

  - In Sect. 3.2.3 (l. 206–208): "[...] Pixels flagged as surfaces or features are not considered in the averaging window. However, if the center pixel of the averaging window (i.e. the pixel to which the averaging is applied) is a low confidence pixel (i.e. "Likely artifact", (A)FA, or "Low confidence small strips"), then the averaging window is applied and, if the average signal value exceeds the detection threshold, this center pixel in the feature detection mask is "unflagged" until the end of the detection level processing, after which its low confidence flag is restored. This allows us to maintain connections between features separated by a few low confidence pixels. Once the averaging is performed, the detection substeps (Sect. 3.2.1) are then applied to the averaged signal. Note too that horizontally adjacent  features  separated only by a low confidence vertical band (i.e., pixels classified as FA, AFA, and/or small strips)  are considered as a  single, merged feature when counting the number of connected pixels $n$. Some examples of this horizontal merging are seen in the smaller fragments of the aerosol layer found at about 4 km and an along track distance of 500 km to 750 km in Fig. 7.".

- Page 10, line 205: 'faint layers is mainly in the horizontal direction'. Please elaborate! Can you specify this more, i.e. aerosol and thin cirrus regions have a long horizontal scale, for aerosols this can go up to 50-100 km. Also the horizontal window allows for the detection of thin layers close to each other vertically

- **Response:** We agree with the reviewer. Thank you.

- **Manuscript modification:**

  - In Sect. 3.2.3 (l. 205): "We chose a horizontal window here because the spatial extent of very faint layers is mainly in the horizontal direction. Typically, thin cirrus have geometrical thicknesses of a few hundreds of meters but spread horizontally over several hundreds of kilometers. The use of a horizontal averaging window thus allows the detection of thin layers close to each other vertically.".

- Page 12 line 241: 'yellow and white colors do not discriminate aerosol from cloud, as in the VFM, but instead simply differentiate weak from strong features based on whether the feature detection required data averaging (yellow) or not (white).' This is actually a very important feature of these kind of masks and it is mentioned here for the first time explicitly. You need to mention this separation on strong and weak instead of cloud & aerosol in the abstract, introduction and conclusion as it is conceptually very different from the standard methods that you only look for non-molecular and not the type.

- **Manuscript modification:**

  - In Introduction (l. 7–9): "Because the algorithm looks for  contiguous 2D patterns using successively lower detection thresholds, it reports strongly scattering features separately from weakly scattering features thus potentially offers improved discrimination of juxtaposed cloud and aerosol layers.".

  - In Conclusions (l. 260–261): "Ideally, separate identification of strongly scattering and weakly scattering features by the 2D-McDA  will also offer improved discrimination between juxtaposed cloud and aerosol layers or identifiable regions of ice and liquid water within a cloud.".

- Page 13 line 283-286 : 'While .... algorithms'. You are very correct with the future outlook that you would like a 2D feature typing in the end, however there are lower hanging fruits to pick first. The detected features will improve and guide the smoothing strategies of the data. This will already show large improvements in the retrievals, i.e. strong and weak features are separately smoothed to calculate the local profile and provide you with the best possible lidar, color and depol. ratio's to enable a better typing and aerosol sub-typing. I think a somewhat broader discussion on how your product can be used is needed here.

- **Response:** Many thanks for your comment. However, at present we are not creating a "product". Instead, we have created a new algorithm that, we hope, will eventually be the foundation for several new products, including a complete redesign of the CALIPSO data products. As you suggest, we have now begun exploring how best to use the enhanced information gleaned from the 2D-McDA to develop a companion set of scene classification algorithms that will "provide [us] with the best possible lidar, color and depol. ratios to enable a better typing and aerosol sub-typing". But as we are still in the early stages of our investigations, at this point we prefer not to speculate in print on the magnitude of the improvements we can achieve.

- **Manuscript modification:**

  - In Conclusions (l. 283–286): "However, while fundamentally important, feature detection is only the first step in extracting comprehensive suite of geophysical parameters from raw lidar measurements. Taking full advantage of the improved spatial analyses delivered by the 2D-McDA thus requires the development of a companion set of 2D scene processes to replace the 1D profile-based processes that are currently used in the CALIPSO retrieval architecture to perform the essential tasks of discriminating between clouds and aerosols, identifying cloud thermodynamic phase, and classifying aerosols by type.".